# ACC-Collab: An Actor-Critic Approach to Multi-Agent LLM Collaboration

**Andrew Estornell** *
ByteDance Research
andrew.estornell@bytedance.com

**Jean-François Ton** *
ByteDance Research
jeanfrancois@bytedance.com

**Yuanshun Yao** †
Meta GenAI
kevinyao@meta.com

**Yang Liu** †
University of California, Santa Cruz
yangliu@ucsc.edu

## ABSTRACT

Large language models (LLMs) have demonstrated a remarkable ability to serve as general-purpose tools for various language-based tasks. Recent works have demonstrated that the efficacy of such models can be improved through iterative dialog between multiple models. While these paradigms show promise in improving model efficacy, most works in this area treat collaboration as an emergent behavior, rather than a learned behavior. In doing so, current multi-agent frameworks rely on collaborative behaviors to have been sufficiently trained into off-the-shelf models. To address this limitation, we propose ACC-Collab, an **A**ctor-**C**riti**c** based learning framework to produce a two-agent team (an actor-agent and a critic-agent) specialized in collaboration. We demonstrate that ACC-Collab outperforms SotA multi-agent techniques on a wide array of benchmarks.

## 1 INTRODUCTION

Recently, large language models (LLMs) have rapidly become a cornerstone in various applications, redefining how we process and generate language at scale (Thirunavukarasu et al., 2023; Hadi et al., 2023; Jiang et al., 2024). Their ability to handle diverse tasks, from translation (Zhu et al., 2024; Otter et al., 2020) to answering complex questions (Zhang et al., 2024; Hao et al., 2024; Havrilla et al., 2024), has attracted the attention of both industry as well as academia. However, despite these advancements, LLMs still exhibit notable weaknesses, particularly when it comes to answering factual questions and reasoning (Tonmoy et al., 2024; Rawte et al., 2023; Huang et al., 2023).

To address these limitations, several techniques have been proposed, such as Chain-of-Thought (CoT) prompting (Wei et al., 2022), Self-Reflection (Ji et al., 2023; Shinn et al., 2023), and Multi-Agent Debate (MAD) (Du et al., 2023), to name a few. These approaches aim to improve the reasoning abilities of LLMs by guiding them toward more accurate answers through structured thinking or discourse. However, the majority of these techniques do not involve training the model specifically for these tasks but instead rely on zero-shot or few-shot capabilities.

Similar to most multi-agent paradigms, MAD approaches make use of off-the-shelf general-purpose LLMs, which are not trained to collaborate. Such approaches rely on collaboration as an emergent, rather than a learned, behavior. While, in some cases, these emergent behaviors are sufficient, the question remains: Can these methods be improved by imbuing models directly with collaborative abilities? To answer this, we propose training teams of LLMs to solve tasks collaboratively.

A particularly relevant work is DebateGPT (Subramaniam et al., 2024), which employs debate as a mechanism to generate higher-quality fine-tuning data. Unlike our approach, which optimizes LLMs for multi-round collaborative problem-solving, their method focuses on using debate to enhance training data for a single model that produces individual responses.

---

*Equal contribution. Correspondence to {andrew.estornell,jeanfrancois}@bytedance.com
†Work done while at ByteDance Research
Code available at https://github.com/LlenRotse/ACC-Collab

In this paper, we propose a novel framework **A**ctor-**C**riti**c** Collaboration (ACC-Collab) which jointly trains a two-agent team to collaboratively solve problems through iterative conversation; this team consists of an actor-agent, responsible for providing answers for a given task, and a critic-agent, responsible for assisting the actor-agent with feedback on its answers. In our training pipeline, we introduce a novel off-policy learning scheme called "Guided-Collaboration" to generate high-quality multi-turn training data to enhance the actor's and critic's performance on challenging tasks.

To summarize, our contributions are as follows:

- We are the first to propose a framework for jointly training a team of LLM agents (Actor-Critic) within the context of collaborative problem solving.

- We introduce a novel data generation scheme, "Guided Collaboration Trajectories", which enables the efficient creation of high-quality training data for both the actor and critic roles.

- Our extensive experiments demonstrate that our method, ACC-Collab, significantly outperforms existing state-of-the-art approaches.

## 2 RELATED WORK

Our research is closely related to the emerging field of multi-agent deliberation, sometimes called Multi-Agent Debate (MAD), which examines how to use groups of models to solve tasks through iterative discussion Chan et al. (2023); Liang et al. (2023); Du et al. (2023); Li et al. (2023c); Khan et al. (2024); Michael et al. (2023); Rasal (2024); Pham et al. (2023); Abdelnabi et al. (2023); Hong et al. (2023); Irving et al. (2018); Li et al. (2023b;d; 2024a); Wang et al. (2023a); Zhang et al. (2023). Many of these works find that language models have naturally collaborative abilities Singhal et al. (2023); Du et al. (2023); Chan et al. (2023), while others have noted that the collaborative ability of off-the-shelf models can be quite limited Wang et al. (2024); Smit et al..

Current approaches to multi-agent deliberation can be broadly cast into two main categories: those that modify model prompts and responses during the discussion Liang et al. (2023); Khan et al. (2024); Rasal (2024); Feng et al. (2024); Yang et al. (2024), and those that modify the structure of the deliberation process Li et al. (2023a); Hong et al. (2023); Liu et al. (2023); Li et al. (2024c); Wang et al. (2023b); Wu et al. (2023); Chen et al. (2023); Chang (2024b). Importantly, both categories use off-the-shelf language models (which have not been trained to collaborate) and work by modifying either the inputs or outputs of these models. Deviating from this line of work, we aim to specifically train a team of models to collaboratively solve tasks.

Two works of particular note are that of Subramaniam et al. (2024), which proposes to use debate data to fine-tune models, and Li et al. (2024b), which trains models for adversarial debate. In the former, debate is used to generate higher-quality fine-tuning data and is not used at inference time; differing from this work, we train models directly to collaborate and use multi-agent discussion both during training and inference. In the latter, models are trained to be effective arguers rather than collaborators, i.e., models are trained to give conceiving arguments such that they can *win* a debate against other LLMs. Differing from this work, we train models to collaboratively solve tasks.

In the context of multi-agent deliberation, the concept of *divergent opinions* is highly relevant to our method. Several approaches to multi-agent deliberation aim to control the level of disagreement among the agents Liang et al. (2023); Khan et al. (2024); Chang (2024a). Often, these works dynamically increase disagreement to prevent early convergence of deliberation. In our study, we leverage divergent opinions to generate high-quality training data. In particular, we have agents change their opinion during the discussion and measure whether or not that change increases or decreases the likelihood that the agents' discussion converges to a correct answer. Using this signal we can then asses the *value* of a given training example for training the models.

Also closely related to our work are paradigms that aim to use self-generated data to improve model performance, often in the context of reasoning or chain of thought Trung et al. (2024); Huang et al. (2024); Xiong et al. (2024); Chen et al. (2024); Pang et al. (2024b). Similar to this line of research, we make use of model generations as training data. However, we are the first work to use such data in the context of multiple models debating collaboratively to solve a given task.

## 3    PRELIMINARIES AND NOTATION

In this section, we formalize multi-agent collaboration between an Actor (an agent that provides answers) and a Critic (an agent that provides feedback to the actor) while also introducing the notation that will be used throughout the remainder of the paper.

Let $(x, y) \sim \mathcal{D}$ be a task-answer pair source from a distribution of tasks and answers $\mathcal{D}$. For a given task $x$, two agents – an actor agent responsible for providing answers and a critic agent responsible for providing feedback and assistance to the actor agent – engage in an iterative discussion over $T$ rounds, to correctly infer the answer $y$. Let $\theta_a$ and $\theta_c$ be the parameters of actor and critic agent, respectively. The iterative discussion between these two agents is as follows:

1. At round $t = 0$ a task $x$ is given to the actor $\theta_a$ who provides an initial response $z_a^{(0)}$.

2. Next, still at round $t = 0$, the critic $\theta_c$ views task $x$ and $z_a^{(0)}$, then provides feedback $z_c^{(0)}$.

3. For each round $t > 0$, the actor views the task $x$, its own previous response $z_a^{(t-1)}$ and the critic's feedback $z_c^{(t-1)}$, then provides an updated response $z_a^{(t)}$.

4. After the actor's new response $z_a^{(t)}$, the critic provides the feedback $z_c^{(t)}$ based on $z_a^{(t)}$.

The accuracy of this procedure is measured via the correctness of the actor's final response, i.e., $\mathbb{I}\left[\zeta(z_a^{(T)}) = y\right]$. Where $\zeta$ is a function that extracts *answers* from text-based responses. For example if $z_a^{(T)}$ = "The sky is blue", then $\zeta(z_a^{(T)})$ ="blue". With this notation and formalization of multi-agent collaboration, we introduce our framework for training actor-critic teams.

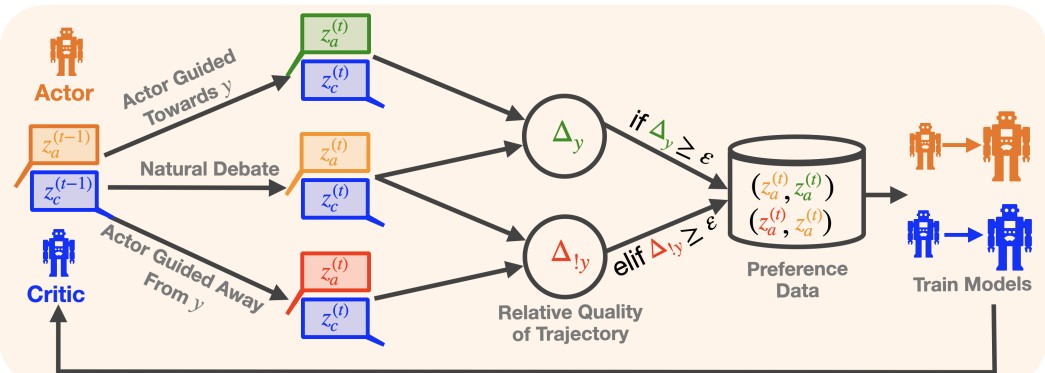

Figure 1: ACC-Collab training pipeline, exemplified for the actor. 1) We generate data from both natural deliberation as well as guided deliberation towards and away from the ground truth answer $y$ using the actor and critic. 2) We compute the relative quality of each trajectory based on the expected quality difference $\Delta_y, \Delta_{!y}$ w.r.t. to the natural response. 3) We store all high-quality pairwise data in our database and train the actor agent. 4) We alternate this procedure for the actor and critic. See Figure 5 of the supplement for the corresponding procedure applied to the critic.

## 4    METHODOLOGY

In this section, we outline our procedure for training a two-agent team, consisting of an actor agent $f_{\theta_a}$ (responsible for providing answers to a given task $x$) and a critic agent $f_{\theta_c}$ (responsible for providing feedback and assistance to the actor). At inference time, the two trained agents engage in iterative discussion to solve a given task $x$, generating the final response $z_a^{(T)}$.

## 4.1 AN ACTOR-CRITIC COLLABORATION FRAMEWORK

Building upon our established notation from the previous section and the general actor-critic framework, we formally define our optimization objective as follows [1]:

$$\theta_a^*, \theta_c^* = \arg\max_{\theta_a} \max_{\theta_c} \mathbb{E}_{(x,y)\sim D}\left[\zeta\left(\underbrace{f_{\theta_a}\left(x, z_a^{(T-1)}, z_c^{(T-1)}\right)}_{\text{actor's final response } z_a^{(T)}}\right) = y\right] \tag{1}$$

Intuitively, Eq. 1 aims to simultaneously optimize the actor's parameters $\theta_a$ and the critic's parameters $\theta_c$, ensuring that the actor's final output at iteration $T$ matches the correct answer $y$. In other words, we optimize the accuracy of the actor's response at time $T$, namely

$$z_a^{(T)} = f_{\theta_a}\left(x, z_a^{(T-1)}, z_c^{(T-1)}\right),$$

where accuracy is measured as $\mathbb{E}\left[\zeta\left(z_a^{(T)}\right) = y\right]$.

It is important to note that the recursive nature of multi-agent deliberation introduces significant complexity to the optimization process. Each response $z_a^{(t)}$ depends not only on the actor's previous output $z_a^{(t-1)}$ but also on the critic's previous output $z_c^{(t-1)}$. This interaction closely resembles a *cooperative dynamic Stackelberg game* (Li & Sethi, 2017), where two players engage in hierarchical decision-making over time, leading us to adopt an *iterative best-response* approach (Fiez et al., 2019). In other words, we first train the critic agent, followed by training the actor to best respond to the critic's output. We can then update the critic to adapt to the newly trained actor, and so on. More formally, this process works by first fixing $\theta_a$, and solving,

$$\theta_c^* = \arg\max_{\theta_c} \mathbb{E}_{(x,y)\sim D}\left[\zeta\left(f_{\theta_a}\left(x, z_a^{(T-1)}, \underbrace{f_{\theta_c}\left(x, z_a^{(T-1)}\right)}_{\text{critic's response } z_c^{(T-1)}}\right)\right) = y\right] \tag{2}$$

then fixing $\theta_c^*$ from above, we solve

$$\theta_a^* = \arg\max_{\theta_a} \mathbb{E}_{(x,y)\sim D}\left[\zeta\left(f_{\theta_a}\left(x, z_a^{(T-1)}, f_{\theta_c^*}\left(x, z_a^{(T-1)}\right)\right)\right) = y\right] \tag{3}$$

this process then repeats until a desired stopping criteria is reached. In practice, we find that a single iteration is sufficient to produce a high-quality collaborative team.

While this alternating scheme allows us to optimize the actor and critic separately, the objectives of each agent still cannot be optimized directly due to the recursive nature of agent responses in this objective; responses at round $T$ depend on those given by the agent at round $t-1$ which themselves depend on the response given at round $t-2$ and so on. To deal with this temporal dependency, we next introduce the concept of Partial Trajectory rewards, which will allow us to capture the signal of each response $z^{(t)}$ for each $t \leq T$.

## 4.2 PARTIAL TRAJECTORY REWARD

To address the inter-round dependencies of the above optimization, we proposed a scheme that allows us to determine the "goodness" of a given response $z^{(t)}$ (from either the actor or the critic) for any $t \leq T$. Consider a conversation between the actor and the critic that was paused at time $t$, i.e., the most recent response is $z^{(t)}$. To assess the goodness of $z^{(t)}$, one might ask *how likely the deliberation procedure will converge to the correct answer $y$ at round $T$*, given that the procedure is already at response $z^{(t)}$. Formally, we can define this as

$$r(z^{(t)}, x, y) = \mathbb{E}\left[\zeta(z_a^{(T)}) = y \mid x, z^{(t)}\right] \tag{4}$$

---

[1]For clarity, we note that the term $\arg\max_{\theta_a} \max_{\theta_b}$ captures the solution for *both* parameters $\theta_a, \theta_c$ in the corresponding bi-level max-max optimization. Here $\theta_c$ is in $z_c^{(T-1)} = f_{\theta_c}\left(x, z_a^{(T-1)}\right)$

Intuitively, the partial reward captures the expectation of arriving at the correct answer $y$ through deliberation starting at round $t$ with generation $z^{(t)}$. In practice, $r(z^{(t)}, x, y)$ can be estimated by learning the reward $r$ or by using heuristics such as one-step roll-out, i.e., Monte Carlo estimation.

In our experiments, we use one-step roll-out heuristics, i.e. simulating an additional deliberation round multiple times from response $z^{(t)}$. The reward $r(z^{(t)}, x, y)$ is set as the average accuracy of these simulations. Empirically, we find this approach effective for generating high-quality training data. We leave learning-based reward functions for future work.

Our objective will then be to optimize the parameters of the actor and critic, $\theta_a, \theta_c$, so that the responses produced by these agents at each timestep $t$, namely $z^{(t)}$, maximize $r(z^{(t)}, x, y)$. That is, we optimize the actor and the critic so that at each timestep $t$, they give a response $z^{(t)}$ which has a high probability of leading the deliberation to converge to the correct answer at time $T$.

To optimize the objective in Eq. 1, we will utilize preference optimization, a standard technique in LLM training. Using the iterative maximization scheme described above, we first have to gather pairwise preference data for both the actor and the critic. In the following sections, we first detail our process for generating this preference data before delving into the optimization procedure.

---

**Algorithm 1:** Trajectory generation and selection

**Data:** Actor and critic: $\theta_a, \theta_c$, Distribution of tasks $\mathcal{D}$, Reward threshold $\varepsilon$
**Result:** A dataset of trajectories $D$
$D \leftarrow \emptyset$ /* Set of trajectories to use */
**for** $(x, y) \sim \mathcal{D}$ **do**
    $\mathbf{z}^{(0)} \leftarrow \text{OneDeliberationRound}(x)$  /* actor and critic responses, i.e. $\mathbf{z} = \langle z_a^{(0)}, z_c^{(0)} \rangle$ */
    **for** $t$ *in* $[1, T]$ **do**
        $\mathbf{z}^{(t)} \leftarrow \text{OneDeliberationRound}(x, \mathbf{z}^{(t-1)})$  /* updated *natural* responses*/
        /* guided-deliberation towards, and away from, correct answer $y$ */
        $\mathbf{z}_+^{(t)} \leftarrow \text{OneGuidedDeliberationRound}(x, \mathbf{z}^{(t-1)}, y)$
        $\mathbf{z}_-^{(t)} \leftarrow \text{OneGuidedDeliberationRound}(x, \mathbf{z}^{(t-1)}, !y)$
        /* Estimate final round accuracy if deliberation continue from response $z^{(t)}$,
        i.e. $r(z^{(t)}, x, y),\ r(z_+^{(t)}, x, y),\ r(z_-^{(t)}, x, y)$ */
        $v \leftarrow \text{EstimateFinalAccuracy}(\mathbf{z}^t)$
        $v_+ \leftarrow \text{EstimateFinalAccuracy}(\mathbf{z}_+^t)$
        $v_- \leftarrow \text{EstimateFinalAccuracy}(\mathbf{z}_-^t)$
        /* Compute the expected improvement for each trajectory */
        /* Save trajectory pairs that result in sufficient accuracy improvement */
        **if** $v_+ - v \geq \varepsilon$ **then**
            $D.\text{add}\big(\text{pos=}\mathbf{z}_+^{(t)}, \text{neg=}\mathbf{z}^{(t)}\big)$
        **end**
        **else if** $v - v_- \geq \varepsilon$ **then**
            $D.\text{add}\big(\text{pos=}\mathbf{z}^{(t)}, \text{neg=}\mathbf{z}_-^{(t)}\big)$
        **end**
    **end**
**end**

---

## 4.3 OFF-POLICY TRAJECTORY GENERATION

In this section, we describe how to generate the preference data needed to optimize the objective in Eq. 1. The classification of a sample as positive or negative is determined by the deliberation trajectory it follows. Specifically, a positive sample for training the actor corresponds to a trajectory likely to lead to the correct answer at round $T$, while a negative sample corresponds to one that leads to an incorrect answer at round $T$. Intuitively, we aim to push the actor agent to generate responses that lead to correct answers while reducing responses that are unlikely to do so, thus optimizing for Eq. 3. The same principle applies to the critic when optimizing Eq. 2.

With this intuition in mind, we now describe how such data is generated. A deliberation trajectory can be defined as a sequence of responses $\langle z_a^{(0)}, z_c^{(0)}, z_a^{(1)}, z_c^{(1)}, \ldots, z_a^{(T)}, z_c^{(T)} \rangle$ for a given task $x$. A straightforward way to generate preference data would be to generate multiple rollouts at each round and select the trajectories with the highest $r(z^{(t)}, x, y)$ as positive samples and those with the lowest $r(z^{(t)}, x, y)$ as negative samples. This approach could enforce the desired behavior for both the actor and the critic if enough samples are collected.

However, this approach is not without its limitations. In particular, if the agent performs poorly on a given dataset, it may be difficult to collect enough positive samples, resulting in low training signals. Additionally, even if the agent performs adequately, generating sufficient responses for both the actor and critic requires significant computational resources, especially to ensure that high $r(z^{(t)}, x, y)$ values are used for positive samples and low values for negative samples.

## 4.4 GUIDED COLLABORATIVE TRAJECTORIES

To address these limitations and improve efficiency, we propose *Guided-Collaborative Trajectories*, which steer the deliberation procedure in two opposing directions: one towards, and another away from, the correct. By comparing these guided trajectories with the natural deliberation trajectory, we can assess the relative *goodness* of each trajectory using an estimation of the reward structure $r$.

Specifically, for task $x$ with answer $y$, let $\mathbf{z}^{(t-1)} = (z_a^{(t-1)}, z_c^{(t-1)})$ be the agents' responses at time $t - 1$. Let $(z_a^{(t)}, z_c^{(t)})$ be the agents' *natural* responses (i.e., without guidance), let $(z_{y,a}^{(t)}, z_{y,c}^{(t)})$ and $(z_{!y,a}^{(t)}, z_{!y,c}^{(t)})$ be the agents responses when guided towards, and away from, supporting answer $y$ respectively. Thus, each guided response is an off-policy generation. In practice, we want guided responses to be different enough from natural responses so that learning the guided responses results in consequential changes to the agent, but not so different that they are challenging to learn; we find that prompt modification is an effective tool for striking this balance. To guide the generations of $(z_{y,a}^{(t)}, z_{y,c}^{(t)})$ and $(z_{!y,a}^{(t)}, z_{!y,c}^{(t)})$, we will simply provide a correct and wrong target answer in the prompt, respectively - see "Guided Collaborative Trajectory Prompts" in Section C for further details.

For each guided response, we consider how influential this response was in altering the accuracy of the final response, i.e., in the case of the actor, we define

$$\Delta_y = r(z_{y,a}^{(t)}, x, y) - r(z_a^{(t)}, x, y) \qquad \text{and} \qquad \Delta_{!y} = r(z_a^{(t)}, x, y) - r(z_{!y,a}^{(t)}, x, y)$$

The terms $\Delta_y$ and $\Delta_{!y}$ give the expected accuracy difference if at round $t$ the actor *had* given response $z_{y,a}^{(t)}$ (or $z_{!y,a}^{(t)}$) instead of response $z_a^{(t)}$. Large $\Delta_y$ indicates that a one-response difference during the deliberation was sufficient to push the procedure toward the correct answer. Such responses would be desirable for the agent to learn. On the other hand, large values of $\Delta_{!y}$ indicate that a one-response difference easily causes the agents to converge to the incorrect answer; this indicates that the deliberation procedure is particularly fragile at timestep $t$.

With these observations in hand, we use $\Delta_y$ and $\Delta_{!y}$ to define positive an negative examples, in particular for a threshold $\varepsilon$,

$$(z_+^{(t)}, z_-^{(t)}) = \begin{cases} (z_y^{(t)}, z^{(t)}) & \text{if} \quad \varepsilon \le \Delta_y = r(z_{y,a}^{(t)}, x, y) - r(z_a^{(t)}, x, y) \\ (z^{(t)}, z_{!y}^{(t)}) & \text{if} \quad \varepsilon \le \Delta_{!y} = r(z_a^{(t)}, x, y) - r(z_{!y,a}^{(t)}, x, y) \end{cases} \tag{5}$$

if neither value is above the threshold, then the example is thrown out.

**Remark 1** *Under Eq. 5, a positive example $z_+^{(t)}$ can be interpreted as a guided response $z_y^{(t)}$ which* increased *the probability of deliberation converging to the correct answer by at least $\varepsilon$, when compared with the natural response $z^{(t)}$. Similarly, a negative example $z_-^{(t)}$ is a guided response $z_{y!}^{(t)}$ which* decreased *the probability of deliberation converging to the answer by at least $\varepsilon$.*

Now that we have a procedure for generating high-quality training examples consisting of positive and negative pairs, we next discuss how to use those positive and negative pairs to train both agents.

### 4.5 LEARNING ON GUIDED TRAJECTORIES

In order to optimize each objective (Eq. 2 and 3), we use standard preference optimization Direct Preference Optimization (DPO) Rafailov et al. (2024). We choose DPO for its efficiency, but any preference optimization scheme could be used (see section B of the supplement for details on how other preference optimization schemes may be incorporated).

Hence, given a preference dataset of positive and negative examples for both the actor and critic agent, of the from $z_-^{(t)}, z_+^{(t)}$, the DPO loss is defined as,

$$\mathcal{L}_{\text{DPO}} = \sum_{t=0}^{T} \mathbb{E}_{(x,y,z_-^{(t)},z_+^{(t)}) \sim D} \left[ \log \sigma \left( \frac{\pi_\theta\left(z_+^{(t)}|x, \mathbf{z}^{(t-1)}\right)}{\pi_{\text{ref}}\left(z_+^{(t)}|x, \mathbf{z}^{(t-1)}\right)} - \frac{\pi_\theta\left(z_-^{(t)}|x, \mathbf{z}^{(t-1)}\right)}{\pi_{\text{ref}}\left(z_-^{(t)}|x, \mathbf{z}^{(t-1)}\right)} \right) \right] \quad (6)$$

Where $\pi_\theta$ is the policy induced by parameters $\theta_a$ or $\theta_c$ and $\mathbf{z}^{(t-1)}$ are the agent's responses at the previous round (i.e., the responses prior to giving either response $z_-^{(t)}$ or $z_+^{(t)}$).

**Remark 2** *Recall, given our generated preference data, this loss implicitly optimizes the reward $r(z^t, x, y)$ which itself is equivalent to final accuracy (i.e., the quantity being maximized by Eq. 1). By summing across all rounds, we implicitly maximize the probability that* each *round $t$ yields a response $z^t$ which causes deliberation to converge to the correct answer at time $T$.*

## 5 EXPERIMENTS

**Benchmarks** To evaluate the efficacy of ACC-Collab we make use of 5 standard benchmark tasks: **BoolQ** Clark et al. (2019) ∼12k yes-no reading comprehension questions, **MMLU** Hendrycks et al. (2020) ∼15k multiple choice questions covering a wide array of subjects and difficulty, **BBH** Suzgun et al. (2022) ∼5k mixed-type questions **SCIQ** Welbl et al. (2017) ∼13k multiple-choice science questions, **ARC** Chollet (2019) ∼7k multiple-choice reasoning-based questions.

**Baselines** We compare ACC-Collab to several multi-agent and single-agent baselines. For inference-based methods, we compare to Society of Minds **SoM** Du et al. (2023), **Persona** Chan et al. (2023). For training-based methods, we compare to supervised fine-tuning **SFT** Radford (2018), **DebateTune** Li et al. (2024b), **DebateGPT** Subramaniam et al. (2024). We use three different base models: Llama-3-8B-Instruct **Llama-3** Dubey et al. (2024), Mistral-7B-Instruct **Mistral** Jiang et al. (2023), and Gemma-2-2B-Instruct **Gemma-2** Team et al. (2024).

### 5.1 ACC-COLLAB PERFORMANCE

**Final Answer Accuracy** We begin by examining the performance of our method ACC-Collab (a single round of training) and ACC-Collab+ (two rounds of training) In table 1, we see the average accuracy of each method after five rounds of deliberation. Our method attains superior performance compared with baseline methods in most cases. The high efficacy of ACC-Collab relative to the baselines indicates that in most cases, only a single round of training is necessary to produce a high-quality collaborative team. It is worth noting that in some cases, further training rounds may decrease performance (i.e., ACC-Collab+ can have worse performance than ACC-Collab). Hence, we have a hold-out set of tasks to determine whether further training degrades performance.

**Process Accuracy** In addition to measuring the correctness of the actor's final answer (i.e., outcome accuracy), we also analyze the correctness of the team's reasoning and discussion steps (i.e., process accuracy). Since ground truth is not available for reasoning and discussion steps, we use GPT-4o as an oracle to evaluate whether the agents follow the correct steps to reach the final answer. We provide an outline of the experimental setup and full results in Section A.1 of the supplement. We find that our method improves or maintains the process accuracy of the actor and critic.

### 5.2 PERFORMANCE INCREASE OF MULTI-AGENT COLLABORATION

**Average Improvement** As noted in Du et al. (2023), the key mechanism behind the success of multi-agent deliberation (or any of its many variants) is that discussion over multiple rounds allows

| | | | | **Llama-3** | | | (Ours) | (Ours) |
|---|---|---|---|---|---|---|---|---|
| | SoM (2x) | SoM (4x) | Persona | DebateTune | SFT | DebateGPT | ACC-Collab | ACC-Collab+ |
| BoolQ | $.812_{\pm.01}$ | $.811_{\pm.007}$ | $.781_{\pm.002}$ | $.775_{\pm.033}$ | $.798_{\pm.006}$ | $.815_{\pm.005}$ | $.887_{\pm.005}$ | $\mathbf{.894}_{\pm.003}$ |
| MMLU | $.62_{\pm.004}$ | $.635_{\pm.004}$ | $.639_{\pm.004}$ | $.63_{\pm.004}$ | $.642_{\pm.005}$ | $.654_{\pm.005}$ | $.644_{\pm.01}$ | $\mathbf{.683}_{\pm.012}$ |
| BBH | $.508_{\pm.003}$ | $.514_{\pm.005}$ | $.509_{\pm.013}$ | $.508_{\pm.005}$ | $.552_{\pm.006}$ | $.551_{\pm.008}$ | $\mathbf{.593}_{\pm.006}$ | $.574_{\pm.003}$ |
| SCIQ | $.925_{\pm.002}$ | $.923_{\pm.002}$ | $.925_{\pm.004}$ | $.924_{\pm.004}$ | $.925_{\pm.003}$ | $.932_{\pm.001}$ | $\mathbf{.952}_{\pm.0}$ | $.948_{\pm.003}$ |
| ARC | $.874_{\pm.001}$ | $.874_{\pm.001}$ | $.87_{\pm.003}$ | $.871_{\pm.002}$ | $.879_{\pm.004}$ | $.876_{\pm.002}$ | $\mathbf{.881}_{\pm.004}$ | $.869_{\pm.002}$ |

| | | | | **Mistral** | | | (Ours) | (Ours) |
|---|---|---|---|---|---|---|---|---|
| | SoM (2x) | SoM (4x) | Persona | DebateTune | SFT | DebateGPT | ACC-Collab | ACC-Collab+ |
| BoolQ | $.801_{\pm.005}$ | $.798_{\pm.004}$ | $.831_{\pm.003}$ | $.83_{\pm.003}$ | $.84_{\pm.003}$ | $.848_{\pm.002}$ | $.877_{\pm.002}$ | $\mathbf{.893}_{\pm.002}$ |
| MMLU | $.57_{\pm.003}$ | $.562_{\pm.005}$ | $.574_{\pm.002}$ | $.562_{\pm.005}$ | $.594_{\pm.004}$ | $.577_{\pm.002}$ | $.61_{\pm.005}$ | $\mathbf{.672}_{\pm.004}$ |
| BBH | $.428_{\pm.002}$ | $.462_{\pm.003}$ | $.465_{\pm.011}$ | $.456_{\pm.005}$ | $.439_{\pm.006}$ | $.48_{\pm.012}$ | $.519_{\pm.009}$ | $\mathbf{.601}_{\pm.004}$ |
| SCIQ | $.856_{\pm.002}$ | $.856_{\pm.002}$ | $.86_{\pm.002}$ | $.863_{\pm.003}$ | $.858_{\pm.004}$ | $.871_{\pm.003}$ | $.902_{\pm.005}$ | $\mathbf{.905}_{\pm.002}$ |
| ARC | $.824_{\pm.001}$ | $.823_{\pm.002}$ | $.827_{\pm.001}$ | $.834_{\pm.0}$ | $.825_{\pm.003}$ | $.822_{\pm.002}$ | $.843_{\pm.003}$ | $\mathbf{.856}_{\pm.003}$ |

| | | | | **Gemma-2** | | | (Ours) | (Ours) |
|---|---|---|---|---|---|---|---|---|
| | SoM (2x) | SoM (4x) | Persona | DebateTune | SFT | DebateGPT | ACC-Collab | ACC-Collab+ |
| BoolQ | $.75_{\pm.011}$ | $.759_{\pm.004}$ | $.716_{\pm.015}$ | $.767_{\pm.003}$ | $.783_{\pm.011}$ | $.812_{\pm.003}$ | $.84_{\pm.005}$ | $\mathbf{.845}_{\pm.005}$ |
| MMLU | $.58_{\pm.002}$ | $.578_{\pm.002}$ | $.577_{\pm.002}$ | $.578_{\pm.001}$ | $.579_{\pm.002}$ | $\mathbf{.582}_{\pm.002}$ | $.51_{\pm.016}$ | $.555_{\pm.003}$ |
| BBH | $.454_{\pm.007}$ | $.449_{\pm.01}$ | $.447_{\pm.006}$ | $.447_{\pm.007}$ | $.498_{\pm.006}$ | $.491_{\pm.01}$ | $\mathbf{.513}_{\pm.006}$ | $.475_{\pm.008}$ |
| SCIQ | $.903_{\pm.002}$ | $.903_{\pm.002}$ | $.908_{\pm.003}$ | $.903_{\pm.001}$ | $.913_{\pm.002}$ | $.914_{\pm.002}$ | $\mathbf{.918}_{\pm.003}$ | $.909_{\pm.003}$ |
| ARC | $.841_{\pm.003}$ | $.843_{\pm.005}$ | $.847_{\pm.003}$ | $.847_{\pm.003}$ | $.848_{\pm.002}$ | $.851_{\pm.003}$ | $\mathbf{.852}_{\pm.003}$ | $.849_{\pm.002}$ |

Table 1: Average accuracy (with 95% confidence intervals) after 5 rounds of deliberation. For each dataset, the highest accuracy is shown in bold.

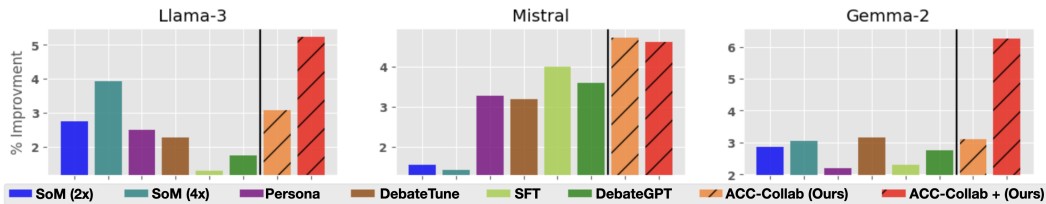

Figure 2: Percent improvement in accuracy after five rounds of deliberation, compared to a single round. Percent improvement (Eq. 7) for each method is averaged across all five datasets.

the models to iteratively refine their answers. Thus, a natural question for any iterative multi-agent method is: *how much does accuracy improve from the initial round $t = 0$ to the final round $t = T$?* Where $T = 4$ in our experiments. To measure this, we look at the percent improvement in model accuracy from round $t = 0$ to round $t = 4$ calculated as,

$$\frac{\text{acc}_4 - \text{acc}_0}{\text{acc}_0} \qquad \text{where acc}_t \text{ is accuracy at round } t \qquad (7)$$

In Figure 2 we see the average percent improvement for each method, averaged across all 5 datasets. For each of the three base models, ACC-Collab+ has the highest average improvement compared to all other methods. Additionally, the improvement gained by methods such as SoM, SFT or DebateGPT is far less stable than that of ACC-Collab+. In particular, for Mistral, SoM yields nearly no improvement, similarly SFT and DebateGPT offer little improvement when applied to of Llama-3.

**Per-Round Accuracy**  Next, we look more closely at the accuracy of each method across five rounds of deliberation. Figure 3, shows per-round accuracy on the BoolQ dataset. As already illustrated by Table 1, ACC-Collab and ACC-Collab+ achieve higher final round accuracy than the other methods. Notably, our method has higher accuracy both at the final round $t = 4$ and at round $t = 0$. Recall that at round $t = 0$, the actor's response is independent of the critic. This indicates that in some cases, our training pipeline can produce actor agents with superior zero-shot accuracy (without deliberation) compared to models produced by SFT and DebateGPT.

Interestingly, we observe that in some cases, SFT and DebateGPT are not much better than simple deliberation with untrained models (i.e., SoM-4x), e.g., BoolQ with Llama-3. This is primarily due to the fact that these methods are designed to improve the model's single-short performance but do

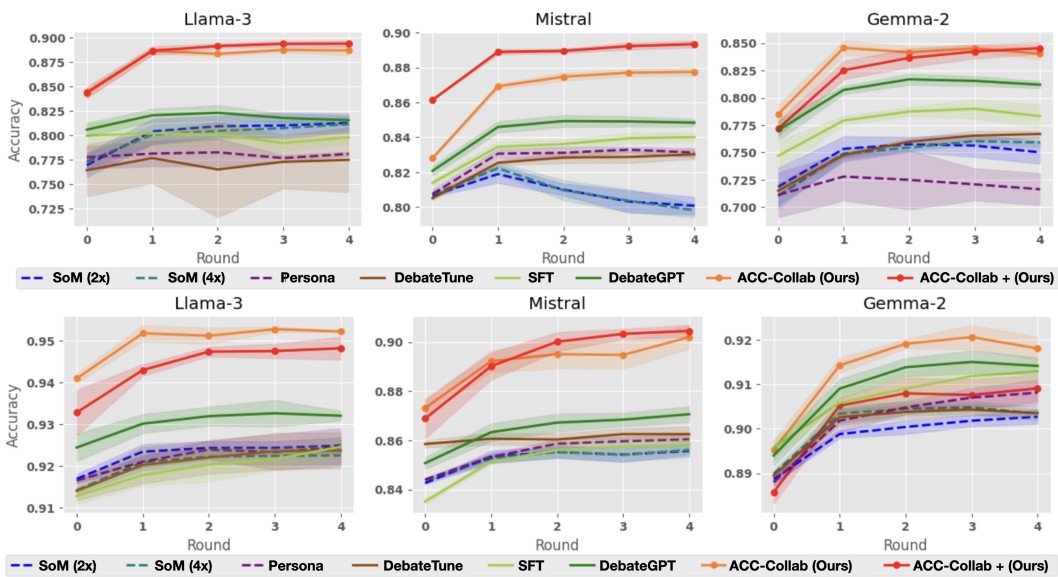

Figure 3: Accuracy over five rounds of deliberation on BoolQ (top) and SCIQ (bottom).

little to improve their collaborative abilities. Cases such as this outline the necessity of training to improve collaboration (ACC-Collab) rather than training for raw accuracy (SFT and DebateGPT).

### 5.3 INDIVIDUAL PERFORMANCE

Next, we examine the relative effectiveness of both the actor and the critic. To do this, we train an actor and critic via ACC-Collab. Then, during deliberation, we pair the trained actor with an untrained critic and pair the trained critic with an untrained actor. In Table 2, column "Actor" corresponds to the former, while "Critic" corresponds to the latter. On average, the trained actor attains higher accuracy compared to the trained critic, this aligns with intuition as the actor is responsible for providing answers while the critic plays a supporting role. In most cases, the trained actor (paired with an untrained critic) outperforms SFT and DebateGPT. When the trained actor is paired with a trained critic (either ACC-Collab or ACC-Collab+), its performance is further improved.

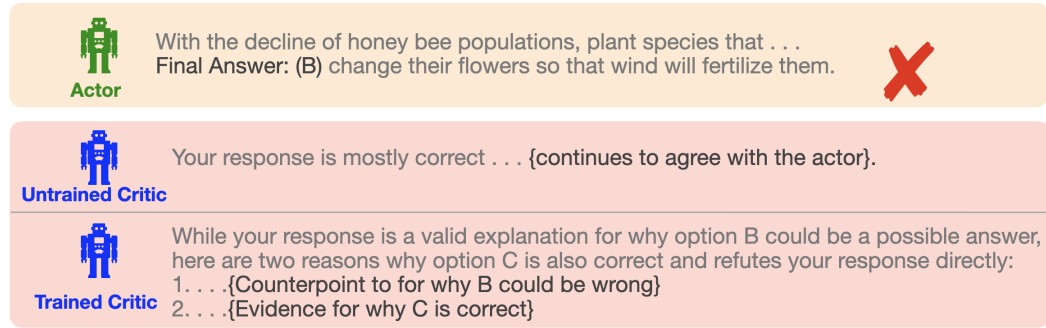

Figure 4: Comparison of responses from the critic model before and after training with ACC-Collab.

### 5.4 WHAT DO THE AGENTS ACTUALLY LEARN?

Lastly, we are interested in understanding how ACC-Collab improves the Actor-Critic Team. Figure 4 demonstrates an example of the difference in responses between an untrained critic and a critic trained through ACC-Collab. Although the actor provides a wrong answer, the untrained critic is too agreeable and does not provide substantive feedback for the actor to correct their answer.

**Llama-3**

|  | SoM (4x) | DebateGPT | Actor | Critic | ACC-Collab | ACC-Collab+ |
|---|---|---|---|---|---|---|
| BoolQ | $.811_{\pm.007}$ | $.815_{\pm.005}$ | $.867_{\pm.003}$ | $.834_{\pm.005}$ | $.887_{\pm.005}$ | $\mathbf{.894}_{\pm.003}$ |
| MMLU | $.635_{\pm.004}$ | $.654_{\pm.005}$ | $.651_{\pm.012}$ | $.65_{\pm.008}$ | $.644_{\pm.01}$ | $\mathbf{.683}_{\pm.012}$ |
| BBH | $.514_{\pm.005}$ | $.551_{\pm.008}$ | $.583_{\pm.01}$ | $.55_{\pm.015}$ | $\mathbf{.593}_{\pm.006}$ | $.574_{\pm.003}$ |
| SCIQ | $.923_{\pm.002}$ | $.932_{\pm.001}$ | $.947_{\pm.002}$ | $.945_{\pm.001}$ | $\mathbf{.952}_{\pm.0}$ | $.948_{\pm.003}$ |
| ARC | $.874_{\pm.001}$ | $.876_{\pm.002}$ | $\mathbf{.885}_{\pm.003}$ | $.866_{\pm.002}$ | $.881_{\pm.004}$ | $.869_{\pm.002}$ |

**Mistral**

|  | SoM (4x) | DebateGPT | Actor | Critic | ACC-Collab | ACC-Collab+ |
|---|---|---|---|---|---|---|
| BoolQ | $.798_{\pm.004}$ | $.848_{\pm.002}$ | $.873_{\pm.002}$ | $.843_{\pm.002}$ | $.877_{\pm.002}$ | $\mathbf{.893}_{\pm.002}$ |
| MMLU | $.562_{\pm.005}$ | $.577_{\pm.002}$ | $.598_{\pm.002}$ | $.611_{\pm.001}$ | $.61_{\pm.005}$ | $\mathbf{.672}_{\pm.004}$ |
| BBH | $.462_{\pm.003}$ | $.48_{\pm.012}$ | $.493_{\pm.012}$ | $.518_{\pm.005}$ | $.519_{\pm.009}$ | $\mathbf{.601}_{\pm.004}$ |
| SCIQ | $.856_{\pm.002}$ | $.871_{\pm.003}$ | $.891_{\pm.001}$ | $.891_{\pm.002}$ | $.902_{\pm.005}$ | $\mathbf{.905}_{\pm.002}$ |
| ARC | $.823_{\pm.002}$ | $.822_{\pm.002}$ | $.833_{\pm.003}$ | $.842_{\pm.003}$ | $.843_{\pm.003}$ | $\mathbf{.856}_{\pm.003}$ |

**Gemma-2**

|  | SoM (4x) | DebateGPT | Actor | Critic | ACC-Collab | ACC-Collab+ |
|---|---|---|---|---|---|---|
| BoolQ | $.759_{\pm.004}$ | $.812_{\pm.003}$ | $.839_{\pm.005}$ | $.774_{\pm.014}$ | $.84_{\pm.005}$ | $\mathbf{.845}_{\pm.005}$ |
| MMLU | $.578_{\pm.002}$ | $\mathbf{.582}_{\pm.002}$ | $.519_{\pm.026}$ | $.566_{\pm.002}$ | $.51_{\pm.016}$ | $.555_{\pm.003}$ |
| BBH | $.449_{\pm.01}$ | $.491_{\pm.01}$ | $.51_{\pm.011}$ | $.475_{\pm.004}$ | $\mathbf{.513}_{\pm.006}$ | $.475_{\pm.008}$ |
| SCIQ | $.903_{\pm.002}$ | $.914_{\pm.002}$ | $\mathbf{.923}_{\pm.002}$ | $.912_{\pm.001}$ | $.918_{\pm.003}$ | $.909_{\pm.003}$ |
| ARC | $.843_{\pm.005}$ | $.851_{\pm.003}$ | $.85_{\pm.002}$ | $\mathbf{.855}_{\pm.002}$ | $.852_{\pm.003}$ | $.849_{\pm.002}$ |

Table 2: Accuracy after 5 rounds of deliberation. The Actor (Critic) column corresponds to an actor-agent (critic-agent) trained via ACC-Collab and paired with an untrained Critic (Actor) during deliberation.

In contrast, the trained critic is more willing to disagree with the actor and provides more detailed feedback. Largely, we observe that this trend is common; untrained critics are too agreeable and are thus less able to change the actor's mind, while trained critics are more willing to disagree (see Section C.2 for more examples). When examining the trained actor's responses, we do not find a notable qualitative change compared to the responses of an untrained actor. However, as shown previously, we do observe a qualitative change in the actor's responses to become more accurate.

## 6 CONCLUSION, LIMITATIONS AND IMPACT

In this paper, we propose ACC-Collab, a novel framework for jointly training a two-agent team (one actor-agent and one critic-agent) to collaboratively solve problems through iterative discussion. To train these agents, we developed an off-policy data generation scheme dubbed "Guided-Collaboration", which produces high-quality preference data for collaborative models. We found that ACC-Collab outperforms all baselines on a wide array of domains. In particular, even a single round of training for both the actor and critic results in a high-quality team. Of particular note is the effects that ACC-Collab has on the critic model. Without ACC-Collab, the critic model is often too agreeable and lacks verbosity in their responses. In contrast, after training with ACC-Collab, the critic is far more likely to provide detailed disagreements during discussion.

However, our framework ACC-Collab also comes with limitations. Firstly, even though ACC-Collab attains superior performance compared to baselines on a wide array of domains, it is important to note that we conduct experiments mainly on question-answering tasks; thus, it remains to be seen whether such a framework would continue to be effective in other types of tasks. Moreover, in our experiments, we train and test models on the same task (partitioning each task into a training and testing set). As such, the generalizability of each actor-critic team to unseen domains is unknown. Our method makes use of the fact that for each question, correct and incorrect answers can be easily established. Secondly, while we provide results for three families of models, these experiments are performed on 2B, 7B, and 8B models. While our method is effective for these sizes (standard in open-source models), it remains to be seen whether this effectiveness will scale to larger models.

ACKNOWLEDGMENTS

We would like to thank Li Hang for his valuable insights and guidance during the development of this work.

REPRODUCIBILITY STATEMENT

Here, we outline the details necessary to reproduce our results. We provide an algorithm for our data generation procedure (Algorithm 1), as well as a description of our training procedure in Section 4.3. Each dataset, baseline method, and base model used are specified at the beginning of Section 5. We provide additional experimental details in Section A. Prompts used for our method can be found in Section C. Lastly, we publicly release the code used for our method.

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

APPENDIX

# A EXPERIMENTS

Here, we provide additional details regarding our experiments.

**Datasets** Each dataset is split into a training set, a validation set, and a testing set. For datasets that come with an explicit partition of these sets we use the given partitions; this includes BoolQ, MMLU, SCIQ, and ARC. For BBH, we randomly sample roughly 25% and 10% of the questions from each category in BBH to create a test and validation set, respectively; this comes out to 1260 questions for the test set and 500 questions for the validation set. All results are reported on questions in the test set.

**Compute** All training was performed on a single Nvidia-H800 GPU. Inference for Llama-3 and Mistral based models is performed on a single Nvidia-v100 GPU, for Gemma-2 based models we used a single Nvidia-H800. All inference is performed with the VLLM library Kwon et al. (2023).

**Training** Training for all models was performed via the trl library, using LoRAs of size 256. When training ACC-Collab with DPO we use a negative log-likelihood (NLL) regularization term (with weight 1) as outlined in Pang et al. (2024a).

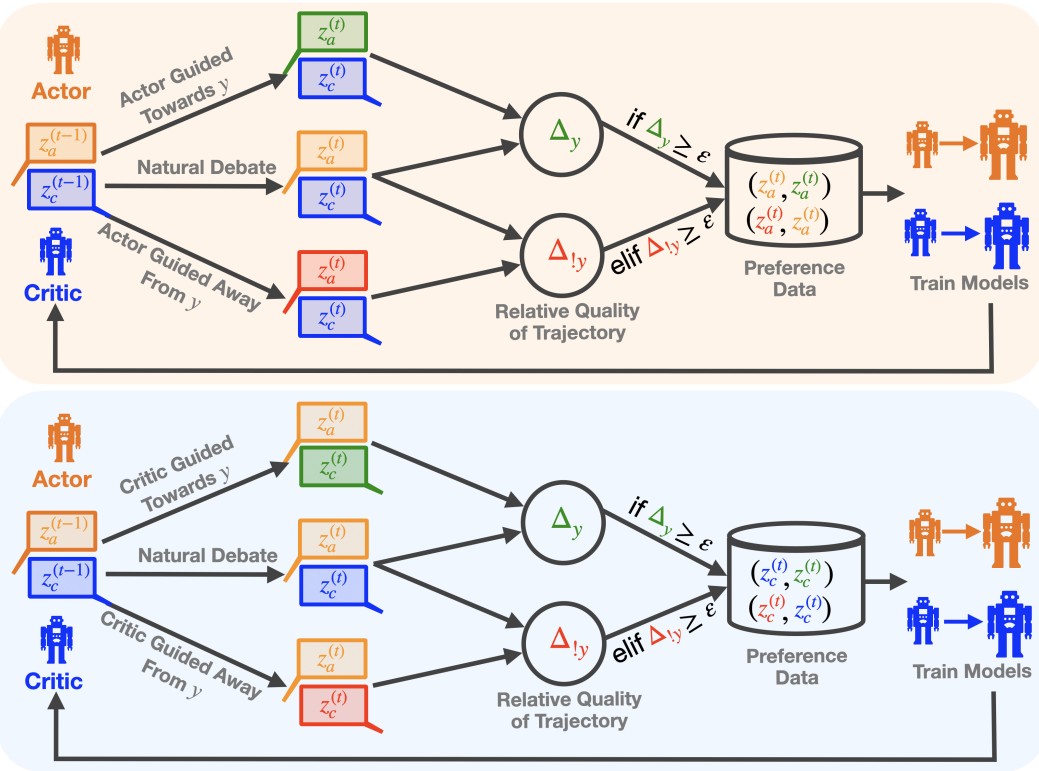

Figure 5: ACC-Collab training pipeline, exemplified for the actor (top) and critic (bottom). The process 1) We generate data from both natural deliberation as well as guided deliberation towards and away from the ground truth answer $y$ using the actor and critic. 2) We compute the relative quality of each trajectory based on the expected quality difference $\Delta_y, \Delta_{!y}$ w.r.t. to the natural response. 3) We store all high-quality pairwise data in our database and train the actor model. 4) We alternate this procedure for the actor and critic. As outline in Section 4 both the guidance procedure and the computation of $\Delta_y, \Delta_{!y}$ differ between the actor and critic.

A.1 CORRECTNESS OF REASONING STEPS

In addition to measuring outcome accuracy (i.e., the accuracy of the actor's final answer), we also measure process accuracy (i.e., the accuracy of the reasoning and discussion steps taken by the actor and the critic model). These results are displayed in Table 3. We observe that our training pipeline typically maintains or increases the accuracy of the reasoning steps.

Unlike outcome accuracy, process accuracy does not have ground truth answers. Thus we utilize GPT-4o to serve as an oracle. We use the following steps to evaluate the correctness of the reasoning and discussion steps.

1. For both an untrained actor-critic team and a trained actor-critic team (trained via ACC-Collab), we randomly sample 500 deliberations, which resulted in the correct answer after 5 rounds of deliberation.

2. We then prompted GPT-4o to evaluate the correctness of the reasoning and discussion steps taken by the actor-critic teams in each of the 500 deliberations.

3. We then compute the average accuracy of the reasoning and discussion steps as evaluated by GPT-4o.

| Dataset | ACC-Collab+ Accuracy | ACC-Collab Accuracy | Untrained Accuracy |
|---------|----------------------|---------------------|--------------------|
| **Llama-3** | | | |
| BoolQ | $.890_{\pm.035}$ | $.894_{\pm.034}$ | $.794_{\pm.045}$ |
| MMLU | $.968_{\pm.020}$ | $.976_{\pm.017}$ | $.909_{\pm.032}$ |
| BBH | $.519_{\pm.064}$ | $.575_{\pm.055}$ | $.610_{\pm.055}$ |
| SCIQ | $.958_{\pm.022}$ | $.981_{\pm.015}$ | $.937_{\pm.027}$ |
| ARC | $.882_{\pm.046}$ | $.833_{\pm.042}$ | $.883_{\pm.036}$ |
| **Mistral** | | | |
| BoolQ | $.907_{\pm.035}$ | $.928_{\pm.029}$ | $.894_{\pm.034}$ |
| MMLU | $.882_{\pm.042}$ | $.884_{\pm.036}$ | $.878_{\pm.037}$ |
| BBH | $.766_{\pm.045}$ | $.818_{\pm.043}$ | $.538_{\pm.056}$ |
| SCIQ | $.973_{\pm.018}$ | $.972_{\pm.018}$ | $.933_{\pm.028}$ |
| ARC | $.927_{\pm.029}$ | $.952_{\pm.024}$ | $.852_{\pm.040}$ |
| **Gemma-2** | | | |
| BoolQ | $.869_{\pm.038}$ | $.872_{\pm.037}$ | $.861_{\pm.039}$ |
| MMLU | $.844_{\pm.041}$ | $.843_{\pm.041}$ | $.800_{\pm.045}$ |
| BBH | $.597_{\pm.056}$ | $.637_{\pm.054}$ | $.514_{\pm.056}$ |
| SCIQ | $.984_{\pm.014}$ | $.978_{\pm.016}$ | $.985_{\pm.014}$ |
| ARC | $.850_{\pm.040}$ | $.854_{\pm.039}$ | $.834_{\pm.042}$ |

Table 3: Accuracy of the actor's and critic's reasoning and discussion steps as evaluated by GPT-4o.

B PREFERENCE OPTIMIZATION

Here, we remark on how other preference optimization schemes can be used in place of DPO. Broadly speaking, preference optimization schemes can broken into two categories: those which optimize reward directly (e.g., PPO) and those which optimize reward indirectly (e.g., DPO). In the case of the latter, the positive and negative pairs produced by our method (Algorithm 1) can be directly plugged into the preference optimizer.

In the case of explicit reward maximization, the reward function $r(z^{(t)}, x, y)$ can first be learned automatically by simply simulating deliberation. The most straightforward way to do this is to first simulate one full deliberation between the actor and critic, i.e., $\langle (z_a^{(0)}, z_c^{(0)}), \ldots, (z_a^{(T)}, z_c^{(T)}) \rangle$. Then for each $z_a^{(t)}, z_c^{(t)}$ the remaining $T - t$ deliberation steps can be resampled to estimate the corresponding value of $r(z_a^{(t)}, x, y)$ and $r(z_c^{(t)}, x, y)$. These pairs, namely $(z_a^{(t)}, r(z_a^{(t)}, x, y))$ and

$\left(z_c^{(t)}, r(z_c^{(t)}, x, y)\right)$ can then be used to learn the reward function. This reward function can then be plugged into the desired preference optimization scheme.

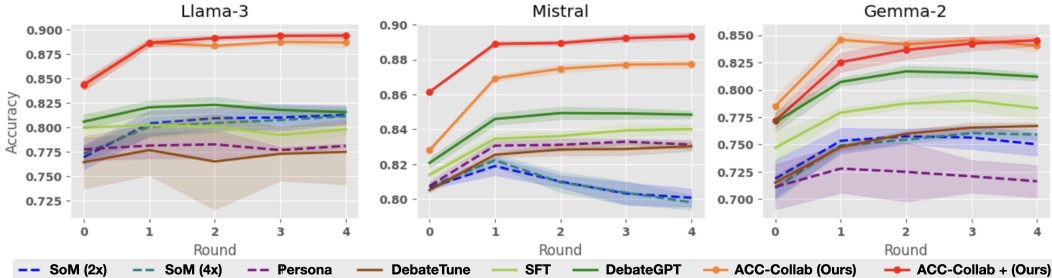

Figure 6: Accuracy over five rounds of deliberation on BoolQ.

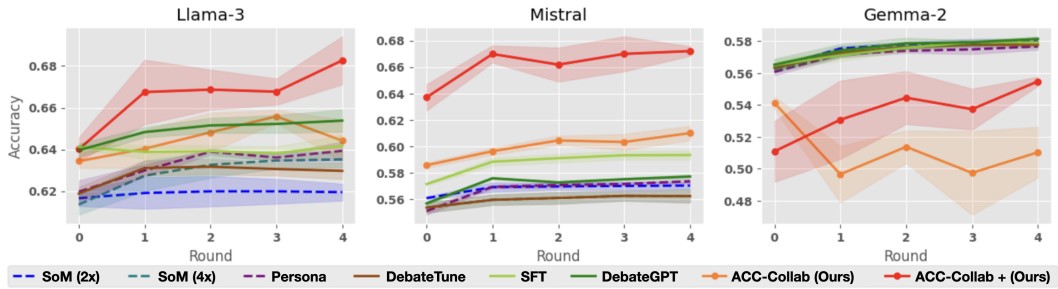

Figure 7: Accuracy over five rounds of deliberation on MMLU.

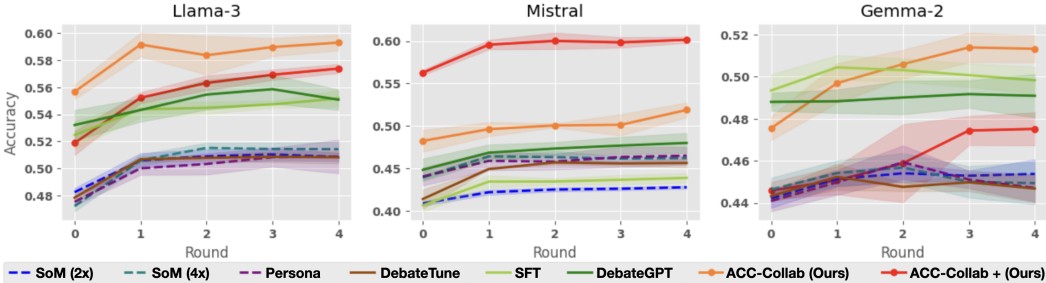

Figure 8: Accuracy over five rounds of deliberation on BBH.

## C  EXAMPLES

### C.1  PROMPTS

Here, we provide examples of the prompts used in our experiments. For illustration, we provide prompts for the BoolQ dataset in which agents are asked a yes-no question about a passage.

**Single-Shot Prompt (No Deliberation)**

```
prompt = ('You will be given a yes-no question which is based on a passage. '
          'You should use the passage to help you answer the question. '
          'You should give a brief justification for your answer, '
          'and you must provide a final answer of either Yes or No.'
          '\nQuestion: {question}?'
          '\nPassage: {passage}'
          )
```

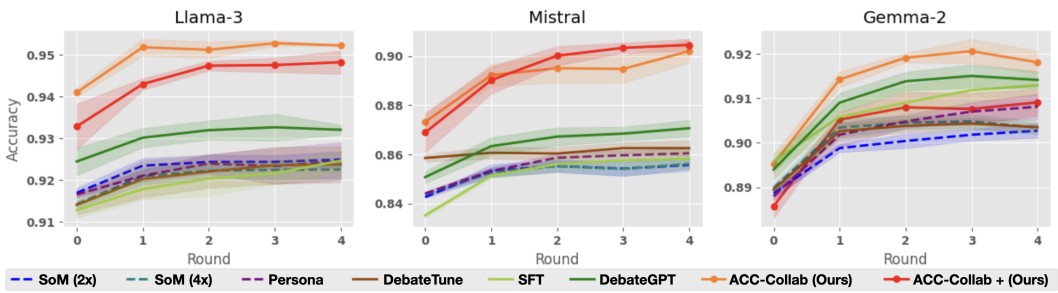

Figure 9: Accuracy over five rounds of deliberation on SCIQ.

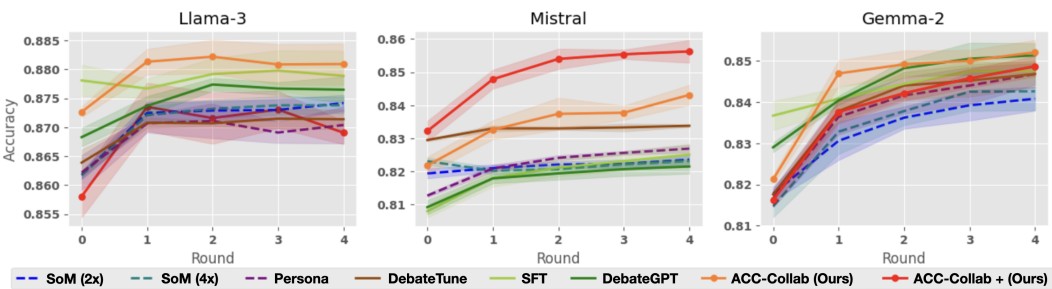

Figure 10: Accuracy over five rounds of deliberation on ARC.

**Guided Single-Shot Prompt (No Deliberation)**

```
prompt = ('You will be given a yes-no question which is based on a passage. '
          'You should use the passage to help you answer the question '
          'with a {target_answer}. '
          'You should give a brief justification for your answer of {target_answer},
           and you must state that your final answer is {target_answer}.'
          '\nQuestion: {question}?'
          '\nPassage: {passage}'
          )
```

**Deliberation Prompt for Actor**

```
prompt = ('Several people have provided answers to a yes-no question. '
          'Below are their responses:'
          '\nPerson {1} said: {responses[1]}'
          '\nPerson {2} said: {responses[2]}'
          .
          .
          .
          '\nPerson {n} said: {responses[n]}'
          '\n\nYou should take these answers into consideration when answering '
          'the following yes-no question, which is based on a passage. '
          'You should give a brief justification for your answer, and you must '
          'provide a final answer of either Yes or No.'
          '\nQuestion: {question}'
          '\nPassage: {passage}'
          )
```

**Guided Deliberation Prompt for Actor**

```
prompt = ('Several people have provided answers to a yes-no question. '
          'Below are their responses:'
```

```
'\nPerson {1} said: {responses[1]}'
'\nPerson {2} said: {responses[2]}'
            .
            .
            .
'\nPerson {n} said: {responses[n]}'
'You should take these answers and the passage into consideration when
'answering the following question with {target_answer}. '
'You should give a brief justification for your answer of {target_answer},
'\nQuestion: {question}'
'\nPassage: {passage}'
)
```

**Deliberation Prompt for Critic**

```
prompt = ('I am answering a question based on a passage. '
          'I would like you to help me improve the correctness of my response '
          'by briefly providing some additional details my original response '
          'may have missed. '
          '\nQuestion: {question}'
          '\nPassage: {passage}'
          '\nMy Response: {actor_response}'
          )
```

**Guided Deliberation Prompt for Critic**

```
prompt = ('I would like you to be a deliberation assistant. '
          'You will be given a question based on a passage and my response '
          'to the question. '
          'You should use the passage and my response to develop additional details
          'for why the correct answer is {target_answer}. '
          'Your details must be brief and must support the fact that the '
          'correct answer is {target_answer}.'
          '\nQuestion: {question}?'
          '\nPassage: {passage}'
          '\nMy Response: {actor_response}'
          )
```

C.2    CRITIC RESPONSES

Here, we present several examples of the critic's responses when the actor provides the wrong answer; critic responses are shown before and after training with ACC-Collab. We observed a qualitative change in the critic's responses after training with our pipeline; the critic was more willing to disagree with the actor, and the critic provided more detailed feedback.

**ACTOR:** [Wrong] Final answer, No. Non-residents cannot bring a handgun into New York for possession or carry within the state based on the passage. They can only transport it through New York to a destination outside of the state where it is legal to possess and carry.

**UNTRAINED CRITIC:** Your response is mostly correct. I would add that New York has strict gun control laws, so even if a non-resident is allowed to transport a firearm through the state, they should be aware that certain areas may have additional restrictions, such as prohibitions on carrying the firearm in certain public places or parks. It's always a good idea to check the specific laws of the jurisdiction they will be passing through and their final destination.

**TRAINED CRITIC:** Details refuting your response:

1. The passage mentions that non-residents can transport any lawful firearm through New York, which implies that the firearm is temporarily in New York state territory during transit. This suggests that under certain circumstances, a handgun could be brought into New York for transit purposes, even if it cannot be carried within the state.

2. The passage does not explicitly state that non-residents are prohibited from bringing handguns into New York. Instead, it only specifies that they cannot possess or carry handguns within the state. This ambiguity leaves room for interpretation that a handgun could be brought into New York for transit purposes as long as it is not carried within the state.

**ACTOR:** [Wrong] Answer: Yes, the passage states that in the Northern Atlantic Ocean, the hurricane season starts as early as June 1 and ends on November 30, with the peak of activity occurring from late August through September. However, the passage also mentions an exception where the hurricane season started as early as January 3 in 1938. Therefore, while the typical start of the hurricane season is June 1, it is possible for the season to begin earlier, as indicated by the passage.

**UNTRAINED CRITIC:** Your response is correct and provides a good justification based on the passage. However, you could add that the passage mentions that May is the least active month for tropical cyclone activity worldwide, but May is still within the Atlantic hurricane season. Therefore, while the peak of the Atlantic hurricane season is from late August through September, the season technically starts earlier.

**TRAINED CRITIC:** While your response is correct in stating that the typical hurricane season in the Northern Atlantic Ocean starts on June 1 and ends on November 30, with the peak of activity occurring from late August through September, the passage also emphasizes that each particular basin has its own seasonal patterns. The passage mentions the exception of the 1938 hurricane season starting as early as January 3, which is outside the typical seasonal window. This detail directly refutes your response by emphasizing that while the typical season starts in June, it is not unheard of for the season to begin earlier, as demonstrated by historical records.

Another detail from the passage that supports the fact that the answer is No, is that the passage states that "worldwide, tropical cyclone activity peaks in late summer," which is contradictory to your response that the hurricane season starts in spring. The passage explicitly states that the peak of tropical cyclone activity occurs in late summer, which is later than spring.

**ACTOR:** [Wrong] Final Answer: (A) rated the group as less interesting and enjoyable.

Justification: Aronson and Mills' study found that severe initiation processes negatively impact immediate attitudes towards the group due to discomfort and negative emotions. (Person 0) Additionally, severe initiation may create pressure to conform and social comparison, making the group less appealing. (Person 1)

**UNTRAINED CRITIC:** Your response is correct. I'd add that Aronson and Mills found that severe initiation rites can create a sense of exclusion and pain, which may negatively impact attitudes towards the group. (A)

**TRAINED CRITIC:** Response: While your justification supports the fact that severe initiation can negatively impact attitudes towards the group, it does not fully refute the alternative answers. Let me provide brief details to argue for the correct answer, B being "rated the group as more interesting and enjoyable":

1. Counter-intuitive findings: Research suggests that negative experiences can sometimes lead to stronger group identification and commitment. The discomfort and negative emotions generated during severe initiation may create a sense of camaraderie and solidarity among group members, ultimately increasing their overall enjoyment of the group (Tajfel & Turner, 1986).

2. Long-term benefits: Although severe initiation might result in negative emotions initially, the feelings of bonding and group cohesion that develop during the initiation process can endure and even enhance members' attachment to the group over time (Sue Carter & Branscombe, 1994). This increased commitment can lead to a heightened sense of enjoyment and engagement with the group.

**ACTOR:** [Wrong] Final Answer: (B) Catalhoyuk and Gobekli Tepe Justification: These sites have provided evidence for social complexity without clear agricultural surpluses. Catalhoyuk had centralized planning and hierarchical social organization but no conclusive evidence for agricultural surpluses. Gobekli Tepe was likely a religious or ceremonial center predating the agricultural revolution.

**UNTRAINNED CRITIC:** Your response is correct. I'd add that Catalhoyuk had a population of up to 8,000 people and was the world's first known urban settlement, while Gobekli Tepe's T-shaped pillars date back to around 9600 BCE and suggest the existence of social and religious institutions that predate agriculture.

**TRAINED CRITIC:** While your justification for option (B) is correct, I would like to provide additional details for option (D) to refute potential responses in favor of other options.

Option (A): Stonehenge and Caral

Justification: Although these sites have evidence of social complexity, there is no clear evidence that they lacked agricultural surpluses. Stonehenge, for instance, is believed to have served as a ceremonial and astronomical center, but its relationship to agriculture remains unclear. Caral, on the other hand, has evidence of irrigation systems, suggesting some agricultural production, even if not a surplus.

Option (C): La Venta and Catalhoyuk

Justification: La Venta is an example of a complex civilization with evidence of agricultural surpluses. While Catalhoyuk is an example of social complexity without clear agricultural surpluses, as you mentioned, La Venta challenges the idea that social complexity always required surpluses.

Option (D): Gobekli Tepe and Poverty Point

Justification: Gobekli Tepe is believed to be a religious or ceremonial center that predates the agricultural revolution. Poverty Point, on the other hand, is a complex society that existed around 3200 BCE and had advanced engineering skills but no clear evidence of agricultural surpluses. These sites' existence undermines the hypothesis that social complexity always required agricultural surpluses, providing further evidence for option (D) being the correct answer.

**ACTOR:** [Wrong] Final Answer: (A) psychological reactance. This is because your friend's statement suggests that they feel frustrated or resistant when external factors prevent them from going on vacation, indicating a desire to have more freedom to make their own choices. Psychological reactance is the tendency to react negatively when our freedom is threatened.

**UNTRAINED CRITIC:** Your response is correct. I'd add that psychological reactance often results in an exaggerated perception of the importance or severity of the obstacles, leading your friend to

believe that there's always a crisis preventing them from vacationing when in reality, the obstacles may not be as significant as they seem.

**TRAINED CRITIC:** While your response is plausible and accurately describes psychological re-actance, let me provide two brief reasons why the correct answer is still (C) illusory correlation:

1. Illusory correlation refers to the perception of a relationship between variables when none or only a weak relationship actually exists. In this case, your friend might perceive a strong correlation between planning a vacation and experiencing crises at home or work, leading them to believe that their vacations are consistently disrupted, even if the actual correlation is weak or non-existent.

2. Psychological reactance is primarily concerned with an individual's desire to react negatively when their freedom is restricted or threatened, not necessarily a misperception of causal relationships. However, the friend's statement does imply a belief that external factors are consistently preventing them from going on vacations, which could be a manifestation of illusory correlation.

**ACTOR:** [Wrong] Final Answer: (D) a zygote. Justification: A zygote is the first cell formed when two gametes merge, marking the beginning of organismic development.

**UNTRAINED CRITIC:** Your response is correct. Just to add, a zygote is formed by the fusion of an sperm (male gamete) and an egg (female gamete), resulting in a diploid cell containing genetic material from both parents. This is the initial stage of an organism's development.

**TRAINED CRITIC:** I agree with your response that the answer is (D) a zygote. However, to further support why (A) a gamete and (C) a nucleus are incorrect, let me provide brief details:

(A) A gamete is a reproductive cell that fuses with another gamete during fertilization to form a zygote. Therefore, it is not an organism in the earliest stages of development but rather a part of the process leading to the formation of an organism.

(C) A nucleus is an organelle within a cell that contains genetic material. It is not an organism but rather a component of a cell, and it is present in the zygote after fertilization has occurred.

So, both (A) and (C) can be ruled out as they do not represent the organism in the earliest stages of development, and the correct answer is (D) a zygote.

**ACTOR:** [Wrong] Final Answer: (A) metamorphic rocks and (B) any types of rocks (specifically igneous rocks when they undergo metamorphosis).

**UNTRAINED CRITIC:** Your response is mostly correct, but it would be more precise to clarify that options B and C also include metamorphic rocks when they mention "any types of rocks" and "igneous rocks," respectively. Metamorphic rocks form from the alteration of pre-existing rocks under heat, pressure, or chemical conditions. Thus, all three options (A, B, and C) are correct, but option A is the most explicit about the transformation process involved.

**TRAINED CRITIC:** Sure! Here are two brief details supporting the correct answer (A) metamorphic rocks and (B) any types of rocks (specifically igneous rocks when they undergo metamorphosis):

1. Metamorphic rocks result from the transformation of pre-existing rocks under high pressure and temperature conditions. This process causes the minerals in the rocks to change, resulting in a new type of rock. For example, sedimentary rocks like shale or limestone can be transformed into metamorphic rocks like slate or marble, respectively.

2. Igneous rocks can also undergo metamorphosis when they are exposed to high pressure and temperature conditions. This process results in the recrystallization of minerals and the formation of new textures. For example, granite can be transformed into gneiss or schist when subjected to metamorphic conditions. Therefore, any type of rock, including igneous rocks, can change and become new types of rocks through metamorphosis.

**ACTOR:** [Wrong] Final Answer: (C) white present

Justification: Gertrude initially had a yellow present, but she swapped it with Alice, who had a white present. Therefore, Gertrude now has the white present.

**UNTRAINED CRITIC:** Your response is correct. I'd add that after Gertrude swaps with Alice, none of the following swaps involve Gertrude giving away or receiving the white present.

**TRAINED CRITIC:** While your justification is correct that Gertrude initially had a yellow present and swapped it with Alice's white present, I would like to add two additional details that refute your response and support option D, brown present.

First, prior to swapping with Alice, Gertrude swapped her yellow present with Dave's brown present.

Second, in the final round of swaps, Fred and Eve swap their presents, and Gertrude's white present (which she received from Alice) is not mentioned in the swap between Fred and Eve.

Therefore, based on the information provided, Gertrude could still have the brown present that she received from Dave earlier in the game.

**ACTOR:** [Wrong] Final Answer: None of the given options are correct. The construction site was closed during the hours Thomas was engaged in the mentioned activities, and there is no information indicating that Thomas visited the construction site before 6am.

**UNTRAINED CRITIC:** Your response is correct. I would add that Thomas' activities from 6am to 8am (gym, yoga studio, and movie theater) all occurred before the construction site opened. Therefore, option (D) can be eliminated as well.

**TRAINED CRITIC:** While it's true that none of the given options are correct, I'd like to provide some brief additional details to support the fact that the correct answer is D (6am to 8am), if that was an option. Here's why:

1. Thomas was seen at the gym from 8am to 11am, and at the yoga studio from 11am to 3pm, meaning he was engaged in those activities between those hours. There's no information indicating that he left those places earlier or returned later.

2. He was seen at the movie theater from 3pm to 6pm, and at the Leaning Tower of Pisa from 6pm to 7pm. Again, there's no information indicating that he left the movie theater earlier or arrived at the Leaning Tower of Pisa later.

3. The construction site was closed after 7pm. However, the options don't include any time before 7pm. If Thomas could have gone to the construction site before 7pm, the correct answer would be one of the options A, B, or C.

4. Since none of the options include the time before 7pm, and Thomas was engaged in activities during the entire day, the only time left for him to visit the construction site, assuming he did, would be before 7am. However, the options don't include that time frame. Therefore, if we had the

option D (6am to 8am), it would be the correct answer, as it's the only time frame that fits the given information.

In conclusion, while the given options are not correct, if option D (6am to 8am) was an option, it would be the correct answer based on the information provided.

