# OpenReview forum: "ACC-Collab: An Actor-Critic Approach to Multi-Agent LLM Collaboration"
_ICLR.cc/2025/Conference — ICLR 2025 Poster_

### Official Review · Reviewer_8Seo · 2024-10-22

**Soundness:** 4
**Presentation:** 2
**Contribution:** 3
**Rating:** 6
**Confidence:** 4

**Summary:**

This paper introduces a new method for training a 2-agent system (Actor & Critic) to better solve tasks collaboratively through debate. This training should provide an improvement over the use of off-the-shelf LLMs which have not been trained with this purpose in mind.
The authors show how the training of this method works and evaluate it on well-known datasets.

**Strengths:**

1. Originality. The paper presents a new framework for improving the multi-agent debate (MAD). This is an interesting direction to generate LLMs that learn to collaborate better.
2. Partial Trajectory Reward is an interesting method to provide reward over the rounds of debate.
3. Robust experimentation and method presentation. The paper presents how the system works and is easy to understand how the training was made.

**Weaknesses:**

1. Minimal improvement. Given the confidence intervals, the improvements achieve after training are not very significant in most cases, when compare to other MAD methods.
2. ACC Debate. The paper presents a method for multi-agent collaboration and mentions the use of Multi-Agent Debate. However, they focused on a the specific case of a Critic dialogue. Even, after reading section 5.3 it is not clear on the impact of using the critic. Unlike this method, the others it's compared with use traditional models of MAD. In my point of view, the kind of debate presented is an actor + critic/judge debate. Therefore, should it also be compared with some other critic system?
3. Generalizability. A note is made for scalability. However, there is no remark on the generalizability of the method. I assume the results are presented for the models trained on the same datasets. What happens when if the model is trained on all of them and later evaluated for each one of them.

**Questions:**

- How do you think that results can be further improved? Can further training or training with more data obtain better results?
- The accuracy improvement of traditional MAD is not very large. But it does provide better factuality. Have you considered using other metric other than accuracy?
- Due to the use of the ACC Debate, this is hardly scalable to be implemented with more agents in the debate, is there a way to do so?

---

> ### Author Response · Authors · 2024-11-19
> **Author Rebuttal (1/2)**
>
> **[Improvement over baselines]**  We thank the reviewer for their interest in our experimental results. However, we believe that there might be a misunderstanding regarding our empirical results. In Table 1 of our paper, we show that our method achieves significant improvements (well outside the confidence intervals) against other MAD methods (both trained and untrained).
>
> In particular, just to point to a few, using the LLama 3 model on the BoolQ dataset, **we are 8.1%, 8.2%, 11.2%, 11.9%, 9.6% and 7.8% better** than SoM(2x), SoM(4), Persona, DebateTune, SFT and DebateGPT respectively. Similarly, for Mistral on the SCIQ dataset, **we are 4.9%, 4.8%, 4.4%, 4.2%, 4.6%, 3.4% better** than SoM(2x), SoM(4), Persona, DebateTune, SFT and DebateGPT respectively. To better illustrate the improvements, we have created the table below to illustrate the **average improvements** of our method compared to existing baselines.
>
> These results illustrate an arguably significant improvement upon baselines and this trend holds across most datasets as well as models. However, we agree that for smaller models (Gemma-2-2b) this is not always the case; our method is equivalent to baselines (SCIQ, ARC), and only worse than baselines for MMLU. We will better highlight these cases in the final version of our paper.
>
> Here positive numbers = our model performed x% better than the baseline and negative meaning our method performed x% worse than the baseline.
> ### Our Method Improvement Over Baselines (Llama-3)
>
> | Dataset | SoM (2x)        | SoM (4x)        | Persona         | DebateTune      | SFT             | DebateGPT       |
> |---------|-----------------|-----------------|-----------------|-----------------|-----------------|-----------------|
> | BoolQ   | 8.1% ± 1.05     | 8.2% ± 0.77     | 11.3% ± 0.37    | 11.9% ± 3.31    | 9.6% ± 0.68     | 7.8% ± 0.59     |
> | MMLU    | 6.3% ± 1.22     | 4.7% ± 1.22     | 4.3% ± 1.22     | 5.3% ± 1.22     | 4.1% ± 1.26     | 2.9% ± 1.26     |
> | BBH     | 8.5% ± 0.45     | 7.9% ± 0.6      | 8.4% ± 1.34     | 8.5% ± 0.6      | 4.1% ± 0.69     | 4.2% ± 0.87     |
> | SCIQ    | 2.7% ± 0.34     | 3.0% ± 0.34     | 2.8% ± 0.49     | 2.8% ± 0.49     | 2.8% ± 0.41     | 2.0% ± 0.29     |
> | ARC     | 0.7% ± 0.22     | 0.7% ± 0.22     | 1.1% ± 0.36     | 1.0% ± 0.28     | 0.2% ± 0.45     | 0.4% ± 0.28     |
>
> ### Our Method Improvement Over Baselines (Mistral)
>
> | Dataset | SoM (2x)        | SoM (4x)        | Persona         | DebateTune      | SFT             | DebateGPT       |
> |---------|-----------------|-----------------|-----------------|-----------------|-----------------|-----------------|
> | BoolQ   | 9.3% ± 0.54     | 9.5% ± 0.45     | 6.2% ± 0.36     | 6.3% ± 0.36     | 5.3% ± 0.36     | 4.5% ± 0.29     |
> | MMLU    | 10.2% ± 0.48    | 11.0% ± 0.62    | 9.8% ± 0.42     | 11.0% ± 0.62    | 7.9% ± 0.54     | 9.5% ± 0.42     |
> | BBH     | 17.4% ± 0.42    | 13.9% ± 0.48    | 13.7% ± 1.16    | 14.5% ± 0.62    | 16.3% ± 0.71    | 12.2% ± 1.26    |
> | SCIQ    | 4.9% ± 0.32     | 4.8% ± 0.32     | 4.4% ± 0.32     | 4.2% ± 0.39     | 4.6% ± 0.47     | 3.4% ± 0.39     |
> | ARC     | 3.3% ± 0.36     | 3.3% ± 0.4      | 2.9% ± 0.36     | 2.2% ± 0.35     | 3.1% ± 0.46     | 3.5% ± 0.4      |
>
> ### Our Method Improvement Over Baselines (Gemma-2)
>
> | Dataset | SoM (2x)        | SoM (4x)        | Persona         | DebateTune      | SFT             | DebateGPT       |
> |---------|-----------------|-----------------|-----------------|-----------------|-----------------|-----------------|
> | BoolQ   | 9.5% ± 1.22     | 8.6% ± 0.67     | 12.9% ± 1.59    | 7.9% ± 0.62     | 6.2% ± 1.22     | 3.3% ± 0.62     |
> | MMLU    | -2.5% ± 0.35    | -2.3% ± 0.35    | -2.2% ± 0.35    | -2.3% ± 0.31    | -2.4% ± 0.35    | -2.7% ± 0.35    |
> | BBH     | 5.9% ± 1.06     | 6.4% ± 1.27     | 6.6% ± 0.99     | 6.6% ± 1.06     | 1.5% ± 0.99     | 2.2% ± 1.27     |
> | SCIQ    | 1.5% ± 0.36     | 1.5% ± 0.36     | 1.0% ± 0.43     | 1.5% ± 0.32     | 0.5% ± 0.36     | 0.4% ± 0.36     |
> | ARC     | 1.1% ± 0.39     | 0.9% ± 0.56     | 0.5% ± 0.39     | 0.5% ± 0.39     | 0.5% ± 0.32     | 0.1% ± 0.39     |
>
> **[Further training to improve results]** We expect that further training iterations will improve the performance of our method. However, as pointed out above and seen in Table 1, our method only requires two rounds of training to outperform baseline methods in all but one case.
>
> **[Metrics other than accuracy]** For each of the baselines that we examine, accuracy is the standard evaluation metric.

---

> ### Author Response · Authors · 2024-11-19
> **Author Rebuttal (2/2)**
>
> **[Comparison with other critic/judge baselines]** We propose a multi-agent debate approach, and as such, we compare our method to two other multi-agent debate methods (SoM and Persona). Additionally, our method uses debate data to train models and we thus compare it to other methods that also use debate data to train models (DebateTune and DebateGPT).
>
> Following the reviewer's suggestion, we have run additional baseline experiments using the critic/judge-based approach in [Int-Debate](https://arxiv.org/pdf/2402.06782). Below we compare our results with SoM (i.e., vanilla debate), Int-Debate, and our two versions of ACC-Debate. We see that typically, Int-Debate does not outperform SoM, and more importantly performs significantly worse than our approach (ACC-Debate) in most cases.
> ### Llama-3
> | Dataset| SoM (2x)       | SoM (4x)       | IntDebate      | ACC-Debate (ours)| ACC-Debate+  (ours)|
> |--------|-------------|----------------|----------------|----------------|----------------|
> | BoolQ  | .812 ± .010    | .811 ± .007    | .814 ± .011    | .887 ± .005    | **.894 ± .003**|
> | MMLU   | .620 ± .004    | .635 ± .004    | .631 ± .006    | .644 ± .010    | **.683 ± .012**|
> | BBH    | .508 ± .003    | .514 ± .005    | .525 ± .005    | **.593 ± .006**| .574 ± .003    |
> | SCIQ   | .925 ± .002    | .923 ± .002    | .930 ± .004    | **.952 ± .000**| .948 ± .003    |
> | ARC    | .874 ± .001    | .874 ± .001    | .871 ± .001    | **.881 ± .004**| .869 ± .002    |
>
> ---
>
> ### Mistral
> | Dataset| SoM (2x)       | SoM (4x)       | IntDebate      | ACC-Debate (ours)| ACC-Debate+  (ours)    |
> |--------|----------------|----------------|----------------|----------------|----------------|
> | BoolQ  | .801 ± .005    | .798 ± .004    | .820 ± .015    | .877 ± .002    | **.893 ± .002**|
> | MMLU   | .570 ± .003    | .562 ± .005    | .565 ± .005    | .610 ± .005    | **.672 ± .004**|
> | BBH    | .428 ± .002    | .462 ± .003    | .456 ± .006    | .519 ± .009    | **.601 ± .004**|
> | SCIQ   | .856 ± .002    | .856 ± .002    | .853 ± .002    | .902 ± .005    | **.905 ± .002**|
> | ARC    | .824 ± .001    | .823 ± .002    | .812 ± .003    | .843 ± .003    | **.856 ± .003**|
>
> ---
>
> ### Gemma-2
> | Dataset | SoM (2x)       | SoM (4x)       | IntDebate       | ACC-Debate (ours) | ACC-Debate+  (ours)    |
> |--------|----------------|----------------|-----------------|----------------|----------------|
> | BoolQ  | .750 ± .011    | .759 ± .004    | .762 ± .016     | .840 ± .005    | **.845 ± .005**|
> | MMLU | .580 ± .002    | .578 ± .002    | **.582 ± .003** | .510 ± .016    | .555 ± .003    |
> | BBH    | .454 ± .007    | .449 ± .01     | .441 ± .007     | **.513 ± .006**| .475 ± .008    |
> | SCIQ   | .903 ± .002    | .903 ± .002    | .901 ± .001     | **.918 ± .003**| .909 ± .003    |
> | ARC    | .841 ± .003    | .843 ± .005    | .848 ± .002     | **.852 ± .003**| .849 ± .002    |
>
> Most works that use critic/judge models only observe superior performance in settings where an external LLM judge is used to evaluate the correctness of a given answer (e.g., QuALITY); two notable works being https://arxiv.org/pdf/2305.19118 and https://arxiv.org/pdf/2402.06782 . Generally, there is little evidence in the literature that critic/judge-based MAD methods are superior to those without critic/judge models, when using untrained models, in the absence of an external judge.
>
> **[Generalizability]** We would likely to clarify that in our experiments, we train and test models on the same task (partitioning each task into a training and testing set). Thus, the generalizability of each actor-critic team to unseen domains is unknown.
> We have added this to our limitation section (in pruple text).
>
> **[Extension to an arbitrary number of agents in debate]** The ACC-Debate pipeline is easily extendable to the case of larger numbers of agents.
> Note that both the reward for partial trajectories $r(z^{(t)}, x, y)$ (line 197) as well as the DPO loss (line 308) are independent of the number of agents in the debate process. As a result, Algorithm 1 can be applied to debate with any number of agents; the only modification required is to change the implementation of the function OneGuidedDebateRound(), which simply runs one round of debate with guided prompts. For simplicity, let's say that we want debate with $n$ actor models (i.e., traditional MAD), then OneGuidedDebateRound() is trivial to implement by simply prompting each of the $n$ actor models with the guided debate prompt provided on line 842.
> Moreover, since our training pipeline uses LoRA fine-tuning, the computational overhead from adding more models is minimal since only one base model needs to be held in memory.
>
> Lastly, we would like to thank the reviewer again for their insightful comments and attention to detail to improve our paper. We hope that the above has addressed any outstanding questions and that the reviewer would consider raising their score if all the questions have been appropriately answered.

---

> > ### Comment · Reviewer_8Seo · 2024-11-20
> > **Rebuttal Comments**
> >
> > Thank you for your comments. I believe the new experiments may add significant value to the paper.
> >
> > I have updated my scores accordingly.

---

### Official Review · Reviewer_j6tQ · 2024-11-04

**Soundness:** 3
**Presentation:** 2
**Contribution:** 3
**Rating:** 5
**Confidence:** 3

**Summary:**

The paper proposes ACC-Debate, an Actor-Critic framework to improve multi-agent debate among LLMs. In traditional multi-agent debates, LLMs often show only emergent collaboration, meaning they aren't specifically trained to work together. The ACC-Debate approach, however, aims to jointly train a team of models (an actor and a critic) to improve reasoning and problem-solving through iterative dialogue.

**Strengths:**

- The paper introduces a new, joint training paradigm for LLMs focused specifically on collaborative problem-solving in debates, unlike previous approaches that rely on emergent collaboration.
- The introduction of the guided-debate data generation approach seems to produce higher-quality training samples, making the learning process more effective and efficient.
- ACC-Debate shows improvements over SoTA debate methods on various benchmarks (BoolQ, MMLU, BBH, SCIQ, ARC).

**Weaknesses:**

- The training pipeline (Figure 1) is unclear. Either in the caption or the paper itself should explain what the green box means, and what the "+" and "-" refers to. Also, from the figure, it seems like ACC-Debate first uses the usual natural debate, then guided debate, followed with training the actor/critic LLMs and iteratively refine the process. I thought the ACC-Debate process only involves guided debate and then iteratively improve the training examples for better training results. Would appreciate some clarifications here.
- Equation 5 seems to be incorrect. Since the authors only observed at most 5 rounds of debate, where $t$ = 0, 1,... 4, why is the percent improvement $\frac{acc_5 - acc_0}{acc_0}$?
- Typo on the y-axis in Figure 2: improvment -> improvement
- What are some error cases of using this ACC-Debate framework, where ACC-Debate fails to improve performance, such as when the actor does not change its response despite the critic’s feedback or when the debate converges on an incorrect answer?
- The authors should conduct ablation studies to examine the effect of the number of debate rounds on final accuracy. This would help determine the optimal number of debate rounds needed for effective debate and identify potential diminishing returns on performance improvement.

**Questions:**

See above weaknesses.

---

> ### Author Response · Authors · 2024-11-19
> **Author Rebuttal**
>
> **[Clarity of Figure 1]** We apologize for the confusion regarding Figure 1. We went through quite a few iterations of this figure, attempting to make it more compact. In hindsight, doing so also removed some important details from the figure. We have included an updated version of this figure in our current draft and updated the PDF on OpenReview. Please let us know if this new version provides more clarity on our method.
>
> **[Clarification on the use of guided and natural trajectories in ACC-Debate]** ACC-Debate uses both guided and natural trajectories. First, we generate natural debate trajectories, which are then used as the starting position for guided trajectories. This is outlined on line 223-227 of Algorithm 1. The natural debate response $z^{(t-1)}$ is used as the starting position for the natural debate trajectory (line 223) as well as both guided trajectories (lines 226 and 227).
>
> **[Typo in Equation 5]** You are correct, this is a typo on our end. The percent improvement in equation (5) should be $\frac{\text{acc}_4 - \text{acc}_0}{\text{acc}_0}$. We have fixed this in the current PDF uploaded to OpenReview.
>
> **[Type of errors made by actor and critic after training with ACC-Debate]** We observe two primary areas of failure in the actor-critic team after training with ACC-Debate. First, as you mention, there are instances where the actor is incorrect but is not persuaded by the critic. Second, we also notice instances in which the critic provides incorrect, but highly persuasive feedback. Balancing the persuasiveness of the critic and the persuadability of the actor is a delicate balance. Despite these occasional errors, we generally observe that ACC-Debate strikes an effective balance (as illustrated by the example in Figure 4).
>
> **[Ablation on accuracy and number of debate rounds]** We agree that examining accuracy as a function of the number of debate rounds is an important evaluation. We would like to highlight that this ablation was provided in Figure 3 (for BoolQ) and Figures 6, 7, 8, 9 (for the other datasets). These figures illustrate that accuracy increases over debate round, up to a point before plateauing. Accuracy typically plateaus after 2 rounds. However, there are some cases in which accuracy continues to increase over all 5 rounds. We note that this relationship between accuracy and debate rounds is common for most debate methods.
>
> We would like to thank the reviewer again for their comments to improve the clarity and quality of our paper. We hope the information provided has resolved any remaining questions; if all concerns have been adequately addressed, we kindly ask the reviewer to consider raising their score.

---

> > ### Author Response · Authors · 2024-12-02
> > **Author Folloup**
> >
> > With the discussion period ending soon, we would like to thank the reviewer again for their time and effort in evaluating our work. If there are any outstanding issues that our rebuttal has not resolved or any additional comments/questions, please do not hesitate to let us know. If our rebuttal has adequately addressed all of the reviewer's concerns, we hope that the reviewer would consider raising their score.

---

### Official Review · Reviewer_yjR9 · 2024-11-04

**Soundness:** 2
**Presentation:** 3
**Contribution:** 3
**Rating:** 6
**Confidence:** 4

**Summary:**

This paper introduces ACC-Debate, a framework that trains two agents, an "actor" and a "critic", to collaboratively solve tasks through structured debate. The framework also incorporates a new off-policy data generation method called "guided debate," which effectively collects positive samples while reducing computational costs. Experimental results show that ACC-Debate outperforms existing debate methods on several benchmarks, indicating that targeted training enhances model collaboration.

**Strengths:**

This paper presents a practical framework for training a team of large language models (LLMs), offering a structured approach to enhance collaborative performance. The proposed method is straightforward and well-defined, making it easily applicable to multi-agent tasks.

**Weaknesses:**

The use of convergence to the correct answer as the reward function for data selection, along with incorporating the correct answer in the prompt for guided debate, may increase the likelihood of false positives (i.e., correct answers reached by flawed processes). This can result in diminished performance after multiple training rounds. The paper would benefit from a thorough analysis of this issue to understand its impact on the robustness of the trained agents. The paper should include an analysis of this issue to understand its impact on trained agents.

**Questions:**

1. Experimental settings are unclear, especially the implementation of Monte Carlo estimation for the reward function.
2. Figure 1 only shows natural debate as a negative sample, inconsistent with Algorithm 1.
3. Equation on Lines 189-190 lacks a reference number.
4. Typo on Line 234: "line improvement" should be "improvement."

---

> ### Author Response · Authors · 2024-11-19
> **Author Rebuttal**
>
> **[Training on guided debate may cause models to reach the correct answer through flawed means]** We do not believe that there is a notable risk of our method causing models to reach correct answers through flawed means. We would expect the guided debate trajectories in ACC-Debate to improve the model's ability to both reach the correct answer as well as to follow correct reasoning/discussion steps to get there.
>
> To verify this hypothesis, we conducted the following experiment for each model and dataset combination:
> - For both an untrained actor-critic team and a trained actor-critic team (trained via ACC-Debate) we randomly sample 500 debates which resulted in the correct answer after 5 rounds of debate.
> - We then prompted GPT-4 to evaluate the correctness of the reasoning and discussion steps taken by the actor-critic teams in each of the 500 debates.
> - We then compute the average accuracy of the reasoning and discussion steps as evaluated by GPT-4.
>
> In the table below, we show the accuracy of our method (ACC-Debate Accuracy) and the accuracy of the untrained actor-critic team (Untrained Accuracy), as well as the difference. We observe that, in nearly all cases, our method attains competitive, and sometimes superior, accuracy compared to the untrained actor-critic team.
>
> ### GPT-4 Evaluated accuracy of reasoning and discussion steps for correct answers
> ### Llama-3
> | Dataset   | ACC-Debate Accuracy | Untrained Accuracy | Difference |
> |-----------|---------------------|--------------------|------------|
> | BoolQ     | 0.894   ±0.034      | 0.794  ±0.045      | 0.100      |
> | MMLU      | 0.976   ±0.017      | 0.909  ±0.032      | 0.067      |
> | BBH       | 0.575   ±0.055      | 0.610  ±0.055      | -0.035     |
> | SCIQ      | 0.981   ±0.015      | 0.937  ±0.027      | 0.044      |
> | ARC       | 0.833   ±0.042      | 0.883  ±0.036      | -0.050     |
>
> ### Mistral
>
> | Dataset   | ACC-Debate Accuracy | Untrained Accuracy | Difference |
> |-----------|---------------------|--------------------|------------|
> | BoolQ     | 0.928   ±0.029      | 0.894  ±0.034      | 0.034      |
> | MMLU      | 0.884   ±0.036      | 0.878  ±0.037      | 0.006      |
> | BBH       | 0.818   ±0.043      | 0.538  ±0.056      | 0.280      |
> | SCIQ      | 0.972   ±0.018      | 0.933  ±0.028      | 0.039      |
> | ARC       | 0.952   ±0.024      | 0.852  ±0.040      | 0.100      |
>
> ### Gemma-2
>
> | Dataset   | ACC-Debate Accuracy | Untrained Accuracy | Difference |
> |-----------|---------------------|--------------------|------------|
> | BoolQ     | 0.872   ±0.037      | 0.861  ±0.039      | 0.011      |
> | MMLU      | 0.843   ±0.041      | 0.800  ±0.045      | 0.043      |
> | BBH       | 0.637   ±0.054      | 0.514  ±0.056      | 0.123      |
> | SCIQ      | 0.978   ±0.016      | 0.985  ±0.014      | -0.007     |
> | ARC       | 0.854   ±0.039      | 0.834  ±0.042      | 0.020      |
>
> We will include this discussion and these experimental results in the supplement of our final version.
>
> **[Clarity of experiments]** When esitmating the reward function for a given trajectory $z^{(t)}$, task $x$, and answer $y$, namely $r(z^{(t)}, x, y)$, we use one-step roll-out with Monte Carlo estimation. This is a common estimation procedure which, despite the name, is amazingly simple. Starting from response $z^{(t)}$, we simulate one additional round of debate multiple times. The average accuracy of these simulations is then the estimated value of $r(z^{(t)}, x, y)$.
> We hope to make our work as clear and well presented as possible. If there are any other areas where you believe clarity can be improved, please let us know.
>
> **[Clarity of Figure 1]** You are correct in that natural debate trajectories can be either positive or negative. Our goal with Figure 1 was to show a minimum complexity version of our pipeline, but we agree that this figure can cause confusion. We have included an updated version of Figure 1 in the main body of our paper and updated the PDF on open review.
> Please let us know your thoughts on this new figure.
>
> **[Typo and equation reference]** Thank you for pointing these out. We have updated these in the current PDF uploaded to OpenReview.
>
> Lastly, we would like to thank the reviewer again for their comments to improve our paper. We hope the information provided has addressed any remaining questions and that, if all concerns have been adequately addressed, the reviewer would consider raising their score.

---

> > ### Comment · Reviewer_yjR9 · 2024-11-26
> >
> > Thank you for your detailed response and the accompanying analysis, which has addressed my initial concerns. I am inclined to raise my score accordingly. However, I believe the issue of false positives may become more pronounced after multiple training iterations, as suggested in lines 345-347 (updated version) regarding ACC-Debate+. Additional experiments on this would further strengthen the paper.

---

> > > ### Author Response · Authors · 2024-11-29
> > > **Author Follow-up**
> > >
> > > > Thank you for your detailed response and the accompanying analysis, which has addressed my initial concerns. I am inclined to raise my score accordingly. However, I believe the issue of false positives may become more pronounced after multiple training iterations, as suggested in lines 345-347 (updated version) regarding ACC-Debate+. Additional experiments on this would further strengthen the paper.
> > >
> > > We agree with the reviewer that as the number of training rounds increases beyond two, the behaviour of the actor and critic is unknown. As a follow-up to your previous suggestion, we have performed the same experiment described above, namely GPT-4 evaluation of the correctness of reasoning/discussion steps in debate, also for ACC-Debate+ (recall that the ACC-Debate is 1 round of training, and ACC-Debate+ is two rounds of training). In the results below we see little difference between the evaluated correctness of ACC-Debate and ACC-Debate+, both of which are consistently evaluated more highly than the untrained models.
> > >
> > > Unfortunately, due to time limitations, it is likely infeasible for us to perform this evaluation for additional training steps. However, we will add this table and associated discussion to the appendix of our paper. Additionally, after thinking more deeply about the concerns raised by the reviewer we will add a remark on this to our limitations section of the final version:
> > >
> > > > However, our framework ACC-Debate also comes with limitations. Even though ACC-Debate attains superior performance compared to baselines on a wide array of domains, it is important to note that we conduct experiments mainly on question-answering tasks; thus, it remains to be seen whether such a framework would continue to be effective in other types of tasks. Moreover, in our experiments, we train and test models on the same task (partitioning each task into a training and testing set). As such, the generalizability of each actor-critic team to unseen domains is unknown. Our method makes use of the fact that for each question, correct and incorrect answers can be easily established. **Since goodness of debate trajectories is evaluated only with respect to these correct and incorrect answers, rather than the process through which these answers are reached, it is possible that models may be trained on incorrect reasoning steps which happened to lead to correct answers; while we do not observe this phenomenon in practice, it is a risk that those wishing to adopt our framework should be aware of.** While we provide results for three families of models, these experiments are performed on 2B, 7B, and 8B models.  While our method is effective for these sizes (standard in open-source models), it remains to be seen whether this effectiveness will scale to larger models.
> > >
> > >
> > > ## GPT-4 Evaluation of Reasoning/Discussion Correctness
> > > ### Llama-3
> > > | Dataset   | ACC-Debate+ Accuracy | ACC-Debate Accuracy | Untrained Accuracy |
> > > |-----------|---------------------|---------------------|--------------------|
> > > | BoolQ     | 0.890   ±0.035      | 0.894   ±0.034      | 0.794  ±0.045      |
> > > | MMLU      | 0.968   ±0.020      | 0.976   ±0.017      | 0.909  ±0.032      |
> > > | BBH       | 0.519   ±0.064      | 0.575   ±0.055      | 0.610  ±0.055      |
> > > | SCIQ      | 0.958   ±0.022      | 0.981   ±0.015      | 0.937  ±0.027      |
> > > | ARC       | 0.882   ±0.046      | 0.833   ±0.042      | 0.883  ±0.036      |
> > > ### Mistral
> > > | Dataset   | ACC-Debate+ Accuracy | ACC-Debate Accuracy | Untrained Accuracy |
> > > |-----------|---------------------|---------------------|--------------------|
> > > | BoolQ     | 0.907   ±0.035      | 0.928   ±0.029      | 0.894  ±0.034      |
> > > | MMLU      | 0.882   ±0.042      | 0.884   ±0.036      | 0.878  ±0.037      |
> > > | BBH       | 0.766   ±0.045      | 0.818   ±0.043      | 0.538  ±0.056      |
> > > | SCIQ      | 0.973   ±0.018      | 0.972   ±0.018      | 0.933  ±0.028      |
> > > | ARC       | 0.927   ±0.029      | 0.952   ±0.024      | 0.852  ±0.040      |
> > > ### Gemma-2
> > > | Dataset   | ACC-Debate+ Accuracy | ACC-Debate Accuracy | Untrained Accuracy |
> > > |-----------|---------------------|---------------------|--------------------|
> > > | BoolQ     | 0.869   ±0.038      | 0.872   ±0.037      | 0.861  ±0.039      |
> > > | MMLU      | 0.844   ±0.041      | 0.843   ±0.041      | 0.800  ±0.045      |
> > > | BBH       | 0.597   ±0.056      | 0.637   ±0.054      | 0.514  ±0.056      |
> > > | SCIQ      | 0.984   ±0.014      | 0.978   ±0.016      | 0.985  ±0.014      |
> > > | ARC       | 0.850   ±0.040      | 0.854   ±0.039      | 0.834  ±0.042      |

---

### Official Review · Reviewer_CH2X · 2024-11-05

**Soundness:** 3
**Presentation:** 4
**Contribution:** 3
**Rating:** 6
**Confidence:** 3

**Summary:**

Multi-Agent Debate (MAD) is an approach to improve the reasoning abilities of LLMs. Different from existing methods that treat debate as an emergent behavior, this paper proposes to treat debate as a learned behavior. It proposes an Actor-Critic based learning framework, which uses reward to find the positive and negative trajectories, and then uses the trajectories to DPO the training preference of the positive sample.

**Strengths:**

1. Authors propose a multi-agent debate method for training LLMs, rather than just utilizing the capabilities of LLMs.

2. The expression of reinforcement learning is clear.

3. The idea of guided-debate is simple but interesting and fits well in DPO training framework.

**Weaknesses:**

The goal of equation 1 is to optimize Actor and Critic by optimizing the two models separately rather than simultaneously, which may result in a failure to achieve the global optimum at the same time. Although it is also mentioned in the paper that PARTIAL TRAJECTORY REWARD solves the dependency relationship between rounds, more theoretical analysis on why equation 1 can achieve global optimum is needed.

**Questions:**

1. The evaluation of r is directly related to the judgment of positive and negative sample trajectories. So how does the reward function learn, and what is its architecture? Is it based on the LLM itself or some other structure?

2. What is your rationale for choosing their current set of benchmarks, and why datasets like GSM8K and MATH were not included?

---

> ### Author Response · Authors · 2024-11-19
> **Author Rebuttal**
>
> **[Optimality and Equation 1]** Equation 1 is an optimization of the actor's accuracy at the final round $T$, and optimizes the actor ($\theta_a$) and the critic ($\theta_c$) jointly. Thus, $\theta_a^*, \theta_c^*$ are definitionally global optima of Equation 1.
>
> Attaining these optimal values is unlikely in practice, as is the case with nearly all deep learning optimization schemes. We do not set out to produce an algorithm that attains optimal values for Equation 1, but rather an algorithm that attains reasonably strong performance in practice. We observe that this is the case as our approach attains higher accuracy than all baselines on all combinations of dataset and model type (with the single exception Gemma-2 on MMLU).
>
> For clarity, we also note that in Equation 1, max-max formulations (i.e., $\max_{\theta_a}\max_{\theta_c}$) are equivalent to joint-max formulation (i.e., $\max_{\theta_a, \theta_c}$).
>
> **[Evaluation of the reward for partial debate trajectories $r$]** As mentioned on line 200, the partial trajectory reward $r(z^{(t)}, x, y)$ can either be learned, or can be approximated via heuristics. In our experiments, we use heuristics to approximate the reward, namely a one-step roll-out with Monte Carlo estimation. This means that we simulate one additional round of debate multiple times (starting from response $z^{(t)}$), the value of $r(z^{(t)}, x, y)$ is then set as the average accuracy of these simulated rounds.
>
> In practice, one could also learn this reward function. However, doing so would introduce additional computation overhead compared with heuristics. We choose to use heuristics as they are more efficient and empirically result in a high-performing debate team; we leave learned reward functions for future work.
>
> **[Choice of benchmarks]** We use 5 standard benchmarks in our experiments and select these benchmarks as they provide a diverse set of tasks. We note that three of our benchmarks (MMLU, BBH, and ARC) contain math and reasoning questions.
> We agree that GSM8K and MATH are common benchmarks in literature, but one can only include so many benchmarks in a single paper, so we elected not to include these benchmarks, relying instead on the fact that MMLU and BBH have math problems.
>
> We hope that we were able to address all the reviewer's questions in the above and hope that the reviewer would consider increasing their score.

---

> > ### Author Response · Authors · 2024-12-02
> > **Author Followup**
> >
> > With the discussion period ending soon, we would like to thank the reviewer again for their time and effort in evaluating our work. If there are any outstanding issues that our rebuttal has not resolved or any additional comments/questions, please do not hesitate to let us know. If our rebuttal has adequately addressed all of the reviewer's concerns, we hope that the reviewer would consider raising their score.

---

### Meta-Review · Area_Chair_19R2 · 2024-12-20

**Metareview:**

This paper proposes a novel method for Multi-Agent debate Using an Actor-Critic approach.  The strengths include a novel method for multi-agent debate, a method for this to happen automatically through reinforcement learning, the guided debate idea which works well with DPO.  The weaknesses include concerns that the method may generate false positives, which should be addressed with more experiments.  Other comments included that the performance improvement was not significant when compared to other MAD methods and that accuracy may not be the best metric for assessing this method.

**Additional Comments On Reviewer Discussion:**

This is truly a borderline paper.  Two reviewers increased their scores from 5 to 6 during the rebuttal phase and two reviewers a 5 and a 6 did not respond to the authors rebuttals.

Overall, having read the paper myself, I could also give it a "6" - I see no reason not to publish it but it could be improved with more work - pushing the publication date out.

I am going to recommend for accept because I believe that scores should have been raised in light of author rebuttal.

---

### Decision · Program_Chairs · 2025-01-22

Accept (Poster)